# Spatiotemporal regulation of autophagy during *Caenorhabditis elegans* aging

Jessica T Chang, Caroline Kumsta, Andrew B Hellman, Linnea M Adams, Malene Hansen*

Program of Development, Aging and Regeneration, Sanford Burnham Prebys Medical Discovery Institute, La Jolla, United States

**Abstract** Autophagy has been linked to longevity in many species, but the underlying mechanisms are unclear. Using a GFP-tagged and a new tandem-tagged Atg8/LGG-1 reporter, we quantified autophagic vesicles and performed autophagic flux assays in multiple tissues of wild-type *Caenorhabditis elegans* and long-lived *daf-2*/insulin/IGF-1 and *glp-1*/Notch mutants throughout adulthood. Our data are consistent with an age-related decline in autophagic activity in the intestine, body-wall muscle, pharynx, and neurons of wild-type animals. In contrast, *daf-2* and *glp-1* mutants displayed unique age- and tissue-specific changes in autophagic activity, indicating that the two longevity paradigms have distinct effects on autophagy during aging. Although autophagy appeared active in the intestine of both long-lived mutants, inhibition of intestinal autophagy significantly abrogated lifespan extension only in *glp-1* mutants. Collectively, our data suggest that autophagic activity normally decreases with age in *C. elegans,* whereas *daf-2* and *glp-1* long-lived mutants regulate autophagy in distinct spatiotemporal-specific manners to extend lifespan.

## Introduction

Macroautophagy (hereafter referred to as autophagy) is a multistep cellular recycling process in which cytosolic components are encapsulated in membrane vesicles and ultimately degraded in the lysosome. Multiple autophagy (*ATG*) genes are involved in the initial formation of a crescent-shaped double-membrane vesicle called the phagophore or isolation membrane (IM), which elongates and engulfs cytosolic cargo, forming the autophagosome (AP). APs then fuse with lysosomes to form autolysosomes (AL), where degradation of cargo takes place (*Mizushima, 2007*).

As interest in this pathway and its pathophysiological roles has increased, it has become clear that measurement of autophagic vesicle levels at steady state, without monitoring the overall pathway flux, can lead to controversial results. Autophagy is commonly monitored by enumerating APs under steady-state conditions, also referred to as the AP pool size (*Loos et al., 2014*), using a GFP-tagged Atg8 marker. During AP formation, Atg8 is cleaved, conjugated to phosphatidylethanolamine, and inserted into the vesicle membrane, thus serving as a marker for IMs and APs (*Klionsky et al., 2016*). However, GFP-Atg8 only reports on the size of the IM and AP pools, not the rate by which IMs and APs are formed, or converted to ALs. For example, an increase in the number of GFP-Atg8 punctae could result from increased formation of APs or blockade of the downstream steps. A tandem-tagged mCherry-GFP-Atg8 reporter, which monitors both IMs/APs (yellow [green/red] punctae) and ALs (red punctae due to GFP fluorescence quenching in the acidic autolysosome environment) can help distinguish between these possibilities (*Kimura et al., 2007*). We acknowledge that APs can also fuse with acidic endosomes and give rise to amphisomes, which are similarly highlighted as red punctae, and which ultimately fuse with lysosomes to form ALs (*Gordon and Seglen, 1988*). Specifically, when used in combination with chemical inhibitors of autophagy, such as

**\*For correspondence:** mhansen@sbpdiscovery.org

**Competing interests:** The authors declare that no competing interests exist.

Bafilomycin A (BafA), tandem-tagged reporters can assess autophagic activity in so called autophagic flux assays (*Klionsky et al., 2016*). Although tandem-tagged Atg8 markers have been used extensively to monitor autophagy in mammalian cells (*Klionsky et al., 2016*), as well as in adult *Drosophila melanogaster* and in *Caenorhabditis elegans* embryos (*Mauvezin et al., 2014*; *Manil-Ségalen et al., 2014*), this reporter has not previously been used in adult *C. elegans*, and no comprehensive spatial or temporal analyses of autophagic activity have been reported in any animal thus far.

Autophagy plays important roles in numerous cellular processes and has been linked to normal physiological aging as well as the development of age-related diseases (*Levine and Kroemer, 2008*). Furthermore, accumulating evidence in long-lived species demonstrates that autophagy genes are required for extended longevity. In particular, autophagy is essential for lifespan extension by inhibition of the nutrient sensor mTOR in *Saccharomyces cerevisiae*, *C. elegans*, and *D. melanogaster* (*Gelino and Hansen, 2012*; *Lapierre et al., 2015*). In *C. elegans*, autophagy genes are also required for the long lifespan induced by other conserved longevity paradigms, such as reduced insulin/IGF-1 signaling, germline ablation, and reduced mitochondrial respiration, and all these longevity mutants have increased transcript levels of several autophagy genes (*Lapierre et al., 2015*). Conversely, neuronal overexpression of *Atg1* or *Atg8* in *D. melanogaster* (*Bai et al., 2013*; *Simonsen et al., 2008*; *Ulgherait et al., 2014*) or ubiquitous overexpression of *ATG5* in mice (*Pyo et al., 2013*) is sufficient to extend lifespan. Consistent with these findings, all long-lived *C. elegans* mutants examined to date display elevated numbers of GFP::Atg8-positive punctae in their hypodermis during larval development and/or in their intestine during early adulthood (*Hansen, 2016*). Long-lived *D. melanogaster* overexpressing Atg1 in neurons also have more GFP::Atg8-positive punctae in neurons and in the intestine than do control animals (*Ulgherait et al., 2014*). Thus, it has been proposed, but not proven, that autophagic activity is elevated in long-lived animals and that this increase is critical for lifespan extension.

Considerable evidence suggests that autophagic activity changes with age, but it is unclear whether increases or decreases in autophagy are causally related to age-associated impairment of cellular function and organismal health. Aging rats have been shown to have reduced lysosomal proteolysis in liver lysates, and reduced autophagic activity in the liver as assessed by electron microscopy and flux assays (*Del Roso et al., 2003*, *Donati et al., 2001*). Moreover, aging is accompanied by a decrease in several autophagy gene transcripts in both *D. melanogaster* (*Demontis and Perrimon, 2010*; *Ling and Salvaterra, 2009*; *Simonsen et al., 2008*) and in rodent tissues (*Cuervo and Dice, 2000*; *Vittorini et al., 1999*; *Ye et al., 2011*; *Kaushik et al., 2012*), as well as with a decrease in lysosomal protease activity in *C. elegans* (*Sarkis et al., 1988*). In contrast, a recent study proposed that autophagic activity is increased in multiple tissues of *C. elegans* with age, as measured by a fluorescently labeled Atg8 reporter protein containing a lysosomal hydrolase-cleavable linker (*Chapin et al., 2015*). However, this steady-state assay did not conclusively evaluate autophagic activity, and the precise step in the autophagic process marked by the reporter was not identified. Taken together, these observations demonstrate a need for additional assays to evaluate and quantify tissue- and age-specific changes in autophagic activity.

To better understand how aging affects autophagy in *C. elegans*, we employed a GFP-tagged and a novel tandem-tagged (mCherry/GFP) form of LGG-1 (a *C. elegans* ortholog of Atg8) to investigate the spatial and temporal autophagy landscape in wild-type (WT) and long-lived *daf-2* insulin/IGF-1 receptor mutants and germline-less *glp-1* animals. To estimate IM/AP (hereafter referred to as AP) and AL pool sizes, we quantified these LGG-1/Atg8 reporters in the intestine, body-wall muscle, pharynx, and nerve-ring neurons between Day 1 and Day 10 of adulthood. We also performed flux assays (i.e. carried out analyses in BafA-injected animals) to assess autophagic activity in each tissue. Our data indicate that WT animals displayed an age-dependent increase in AP and AL numbers in all tissues, which flux assays suggest reflects a decrease in autophagic activity over time. In contrast, *daf-2* and *glp-1* mutants showed unique age- and tissue-specific differences consistent with select tissues displaying elevated, and in one case possibly reduced autophagic activity compared with WT animals. Moreover, tissue-specific inhibition of autophagy in the intestine significantly reduced the long lifespan of *glp-1* mutants but not of *daf-2* mutants, suggesting that autophagy in the intestine of *daf-2* mutants may be dispensable for lifespan extension. Our study represents the first efforts to comprehensively analyze autophagic activity in a spatiotemporal manner of a live organism and

provides evidence for an age-dependent decline in autophagic activity, and for a complex spatio-temporal regulation of autophagy in long-lived *daf-2* and *glp-1* mutants.

## Results

### Spatiotemporal analysis of the GFP::LGG-1 reporter in adult *C. elegans*

To perform an in-depth analysis of autophagy in adult *C. elegans*, we used previously published GFP::LGG-1 reporters (*Meléndez et al., 2003*; *Kang et al., 2007*; *Gelino et al., 2016*) to assess the isolation membrane (IM) and autophagosome (AP) pool size (for simplicity referred to as AP pool size, *Figure 1A*) in multiple somatic tissues of adult WT animals during aging, including the intestine (*Figure 1B*), body-wall muscle (hereafter referred to as muscle; *Figure 1C*), pharynx (the foregut of the animal; *Figure 1D*), as well as nerve-ring neurons (hereafter referred to as neurons; *Figure 1E*). Specifically, we used confocal microscopy and counted the number of GFP::LGG-1 positive punctae in each of these tissues from the first day of adulthood (i.e. when animals become reproductively active) until Day 10 of adulthood (*Figure 1F–I*), at which time GFP::LGG-1-positive punctae could still be identified in these major tissues (*Figure 1B'–E'*). Notably, all tissues examined showed an increase in GFP::LGG-1-positive punctae over time with the muscle showing the largest (~9-fold) increase (*Figure 1F–I*). We confirmed that the GFP::LGG-1-positive punctae likely represented APs and not GFP::LGG-1 aggregates, since WT *C. elegans* expressing a mutant form of GFP-tagged LGG-1/Atg8 protein (G116A) that is expected to be defective in lipidation and autophagosome-membrane targeting (*Manil-Ségalen et al., 2014*) showed no punctae formation in the intestine or muscle, and only a small increase in the pharynx over this 10-day time period (*Figure 1—figure supplement 1*, see figure legend for comment on neurons, which have yet to be evaluated in detail). Taken together, these experiments show that the GFP::LGG-1 reporter can be used to monitor autophagic events in multiple somatic tissues well into old age (i.e. at least until Day 10), and indicate that the AP pool size generally increases over time without forming lipidation-independent aggregates in WT animals. However, these observations are insufficient to evaluate whether the observed increase in GFP::LGG-1-positive punctae represents an induction in the formation, or a block in the turnover of APs.

### Expression of an mCherry::GFP::LGG-1 reporter in adult *C. elegans*

To expand our spatiotemporal analysis, we constructed and expressed a dual-fluorescent mCherry::GFP::LGG-1 protein to monitor both APs as well as autolysosomes (ALs), as originally done in mammalian cells (*Kimura et al., 2007*). With this reporter, APs are visualized as punctae positive for both GFP and mCherry fluorescence, while ALs (and amphisomes, for simplicity referred to here as ALs) emit only the mCherry signal due to quenching of GFP in the acidic environment (*Figure 2A*). Expression of *mcherry::gfp::lgg-1* from the endogenous *lgg-1* promoter produced a full-length mCherry::GFP::LGG-1 protein (*Figure 2—figure supplement 1A–C*) that was functional since it was able to rescue an embryonic lethal *lgg-1(tm3489)* mutant (see Materials and methods, data not shown), as previously shown for GFP::LGG-1 expressed from the *lgg-1* promoter (*Manil-Ségalen et al., 2014*). mCherry::GFP::LGG-1 was expressed in several major tissues of adult WT animals (*Figure 2B–G*, *Figure 2—figure supplement 2A–E*), consistent with the expression profile of GFP::LGG-1 (*Figure 1B–E* and [*Meléndez et al., 2003*]). As previously done for GFP::LGG-1 (*Gelino et al., 2016*), we also expressed mCherry::GFP::LGG-1 from a pan-neuronal *rgef-1* promoter to specifically visualize neurons (*Figure 2G*, *Figure 2—figure supplement 2E*). mCherry/GFP double-labeled and mCherry single-labeled punctae were observed in multiple tissues, including hypodermal seam cells, the tissue most commonly characterized thus far when assessing autophagy in *C. elegans* (*Hansen, 2016*; *Zhang et al., 2015*), the intestine, muscle, pharynx, and neurons of Day 1 adult animals (*Figure 2C–G*, *Figure 2—figure supplement 2A–E*). Immunofluorescence analysis of the intestine of Day 1 mCherry::GFP::LGG-1-expressing animals showed that both GFP- and mCherry-positive punctae co-localized with structures stained by an LGG-1 antibody (*Figure 2—figure supplement 1D–E*), consistent with only full-length protein being present in double-labeled punctae. As observed with the GFP::LGG-1 reporter (*Figure 1B'–E'*), mCherry/GFP and mCherry-only punctae were observed in *mCherry::gfp::lgg-1* transgenic animals at least until Day 10 (*Figure 2D'–G'*, *Figure 2—figure supplement 2B'–E'*). Hypodermal seam cells could not be

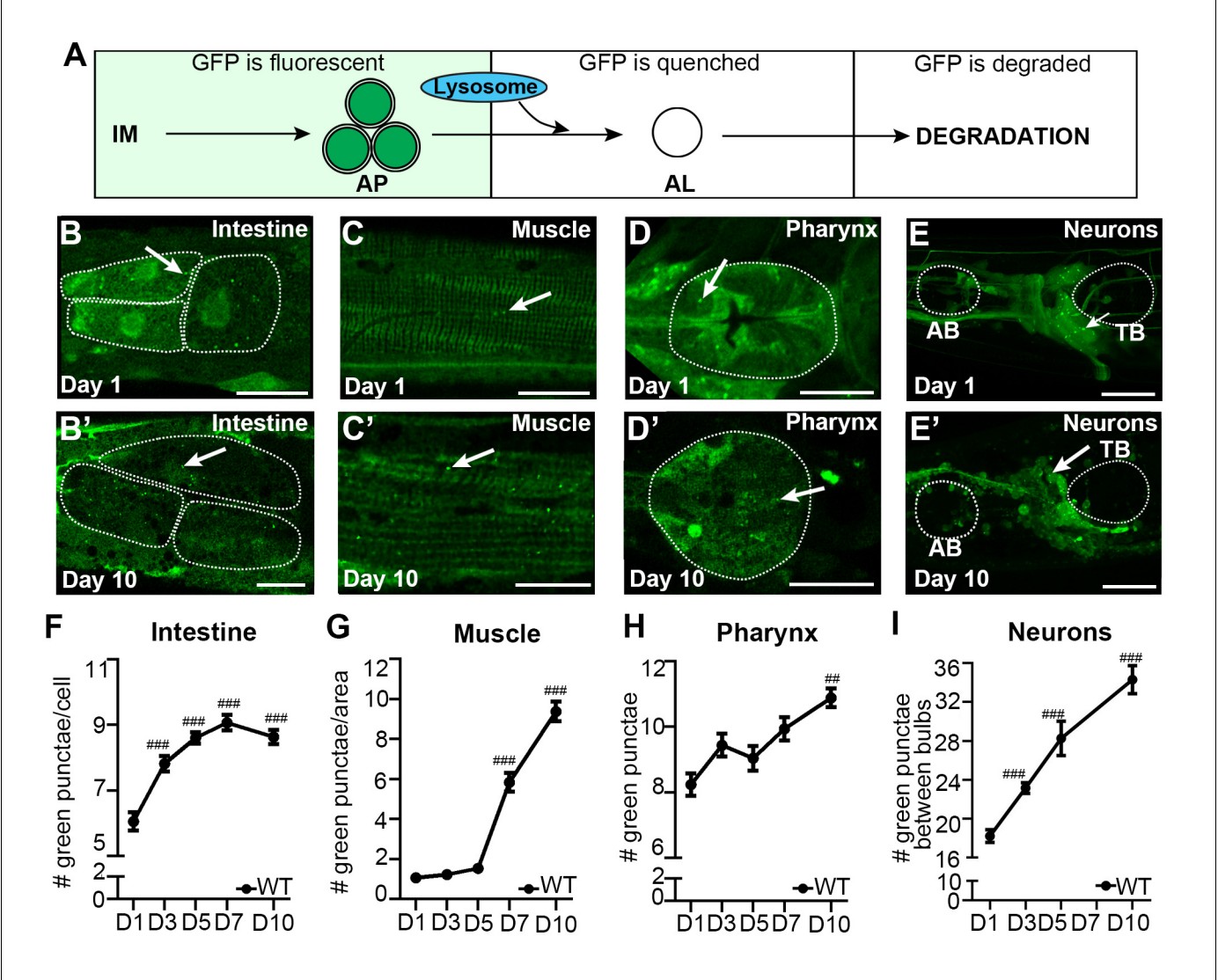

**Figure 1.** The autophagosome pool size increases with age in *C. elegans*. (**A**) Schematic representation of GFP::LGG-1 fluorescence states in the autophagy pathway. IM, isolation membrane; AP, autophagosome; AL, autolysosome. (**B–E'**) Adult transgenic WT animals expressing *gfp::lgg-1*, imaged at Day 1 (**B–E**) and Day 10 (**B'–E'**) of adulthood. APs (arrows) can be seen in the intestine (**B,B'**), body-wall muscle (**C,C'**), pharynx (**D,D'**), and nerve-ring neurons (**E,E'**). Dotted lines outline individual intestinal cells (**B,B'**) and pharyngeal bulbs (**D–E'**). AB, anterior pharyngeal bulb; TB, terminal pharyngeal bulb. Scale bars = 20 μm. (**F–I**) Quantification of autophagosomes (AP; GFP punctae) in the intestine (**F**), body-wall muscle (**G**), pharynx (**H**), and nerve-ring neurons (**I**) at Days 1, 3, 5, 7, or 10 of adulthood in WT animals. Day 7 was omitted for neurons due to a counting issue at this time point. Data are the mean ± SEM of ≥20 animals combined from three independent experiments per time point. ###p<0.00 and ##p<0.001 for WT control at Day 1, 3, 5, 7, or 10 vs. WT control at Day 1 by Poisson regression.

The following figure supplement is available for figure 1:

**Figure supplement 1.** Age-related increase in autophagosome pool size requires lipidation of LGG-1.

visualized in older animals because they fuse and appear as a single line shortly after Day 1 of adulthood. Taken together, these data confirm that transgenic *C. elegans* express an mCherry::GFP::LGG-1 protein that localizes to dual- and single-fluorescent punctae in young and old adults.

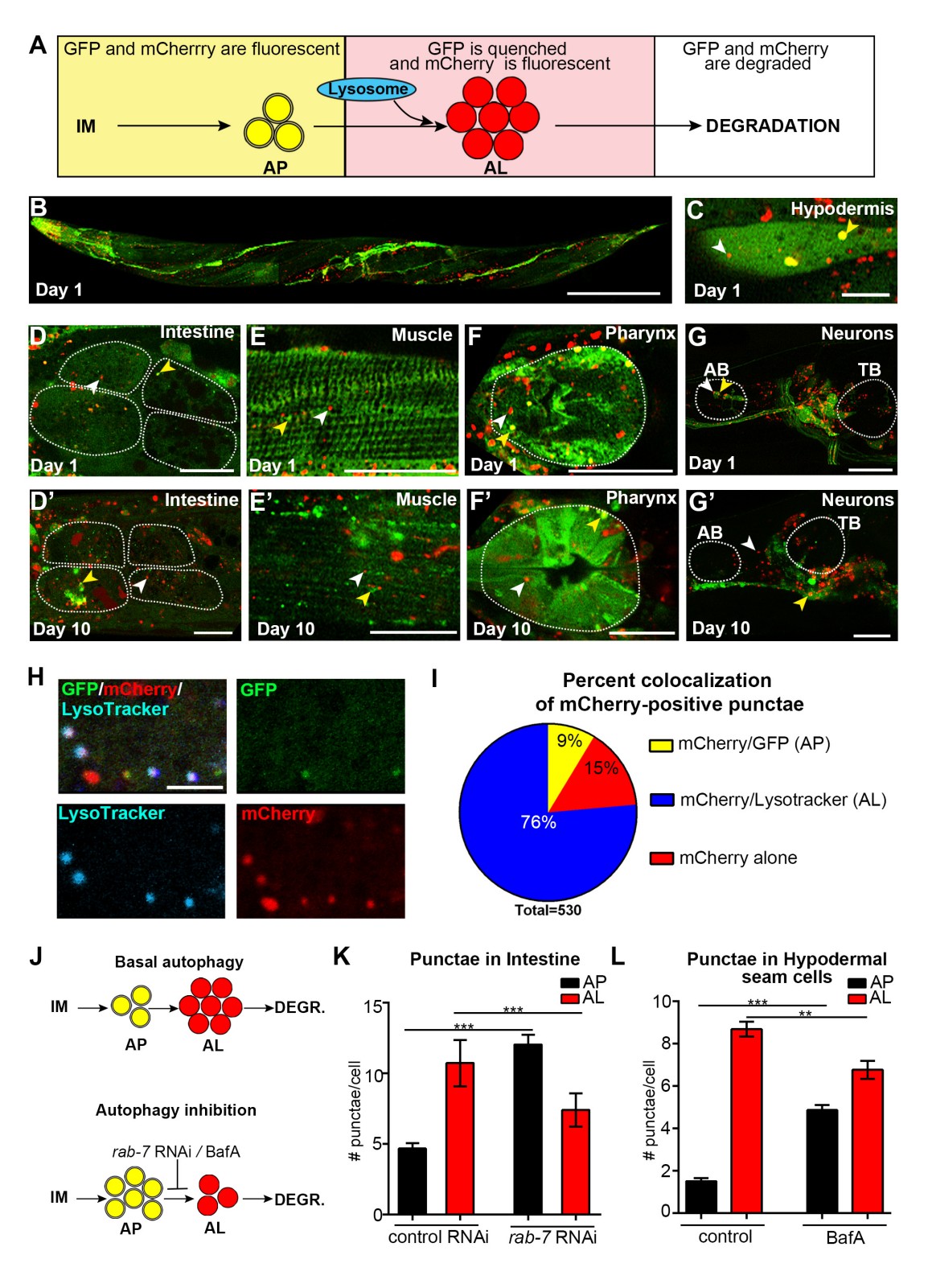

**Figure 2.** Expression and validation of a novel *mCherry::gfp::lgg-1* reporter in *C. elegans*. (A) Schematic representation of mCherry::GFP::LGG-1 fluorescence states in the autophagy pathway. IM, isolation membrane; AP, autophagosome; AL, autolysosome. (B) Whole-body expression of mCherry::GFP::LGG-1 in a wild-type (WT) animal at Day 1 of adulthood. Scale bar = 100 µm. Note that the intensity of red fluorescence compared to green is stronger; thus, the red channel was purposely set lower (see Materials and methods). (C–G') Adult transgenic WT animals expressing *mCherry::*

*Figure 2 continued on next page*

*Figure 2 continued*

*gfp::lgg-1,* imaged at Day 1 (C–G) or Day 10 (D′–G′) of adulthood. APs (mCherry/GFP; yellow arrowheads) and ALs (mCherry only; white arrowheads) can be seen in the hypodermal seam cells (C), intestine (D,D′), body-wall muscle (E,E′), pharynx (F,F′), and nerve-ring neurons (G,G′). Dotted lines outline individual intestinal cells (D,D′) and pharyngeal bulbs (F,G′). AB, anterior pharyngeal bulb; TB, terminal pharyngeal bulb. Scale bars = 20 μm (D–G′) and (C) = 10 μm. (H) Intestine of Day 1 WT transgenic animals expressing *mCherry::gfp::lgg-1* (green, GFP; red, mCherry) and stained with LysoTracker (light blue). Scale bar = 5 μm. (I) Quantification of punctae containing mCherry alone or co-localized with GFP or LysoTracker in the intestine of WT transgenic animals. Data are representative of three independent experiments, each with ≥10 animals. (J) Schematic representation of basal AP and AL pool sizes and the effect of inhibiting autophagy by *rab-7* RNAi or Bafilomycin (BafA) treatment. (K) Quantification of APs and ALs in the intestine of Day 1 WT transgenic animals fed from hatching with bacteria expressing empty vector (control) or dsRNA encoding *rab-7*. Data are the mean ± SEM of ≥40 animals combined from three experiments. ***$p<0.0001$ by Student's *t*-test. (L) Quantification of APs and ALs in the hypodermal seam cells of Day 1 WT transgenic animals injected with BafA or DMSO (control). Data are the mean ± SEM of ≥30 animals combined from three experiments. ***$p<0.0001$ and **$p<0.001$ by Student's *t*-test.

The following figure supplements are available for figure 2:

**Figure supplement 1.** mCherry::GFP::LGG-1 reporter produces a full-length protein.

**Figure supplement 2.** mCherry::GFP::LGG-1 reporter is expressed in multiple tissues.

**Figure supplement 3.** Calculations showing that quantification of autophagic vesicles at steady state provide insufficient information to determine autophagic flux.

**Figure supplement 4.** Additional validation of new mCherry::GFP::LGG-1 reporter.

## mCherry/GFP and mCherry-only punctae represent autophagosomes and autolysosomes, respectively, in mCherry::GFP::LGG-1 transgenic animals

We sought to verify that the mCherry/GFP and mCherry-only punctae observed in mCherry::GFP::LGG-1-expressing animals represented *bona fide* autophagic vesicles. Irrespective of tissue type, the mCherry/GFP-positive punctae (*Figure 2D–G*, *Figure 2—figure supplement 2B–E*) were similar in morphology and number to the GFP-positive punctae observed in transgenic animals expressing GFP::LGG-1 (*Figure 1B–E* and [*Meléndez et al., 2003*]), supporting their identity as APs. To determine whether mCherry-only punctae represent ALs, we examined their co-localization with Lyso-Tracker, a pH-sensitive dye that fluoresces in acidic compartments (*Hersh et al., 2002*). Indeed, ~75% of mCherry-positive punctae in the intestine of Day 1 adult WT animals co-localized with LysoTracker and ~10% co-localized with GFP (*Figure 2H–I*) consistent with labeling of ALs and APs, respectively. Of note, ~15% of mCherry-positive puncta co-localized with neither GFP nor Lyso-Tracker (*Figure 2H–I*). These mCherry-only punctae may reflect inefficient labeling of Lysotracker in the intestine (which was highly variable between animals and experiments, data not shown), or they could represent mCherry aggregates rather than autophagic vesicles; however, we did not observe free mCherry on Western blots (*Figure 2—figure supplement 1B*). While we cannot rule out that this relatively low percentage of mCherry-positive punctae in the intestine could represent aggregates, our data are consistent with the mCherry::GFP::LGG-1 reporter being able to distinguish between APs and ALs in adult *C. elegans*, at least in the intestine of the animal. Hereafter, we will refer to mCherry/GFP and mCherry-only punctae as APs and ALs, respectively. Interestingly, ALs were more abundant than APs in all tissues of WT transgenic animals (*Figure 2D–G*, *Figure 2—figure supplement 2A–E*, *Figure 3*), suggesting that turnover of APs (i.e. formation of APs and conversion to ALs) is faster than turnover of ALs. If so, this would indicate that AL turnover may be rate limiting in *C. elegans* (further addressed in the Discussion; see also *Figure 2—figure supplement 3A–C*).

We further validated the mCherry::GFP::LGG-1 reporter by investigating the effects of genetic (RNAi) or pharmacological inhibition of autophagy on AP and AL numbers. RNAi-mediated inhibition was achieved by feeding *mCherry::gfp::lgg-1* transgenic animals with bacteria expressing dsRNA for *rab-7*, a small GTPase required for AP–lysosome fusion (*Figure 2J* and [*Manil-Ségalen et al., 2014*]). WT *mcherry::gfp::lgg-1* transgenic animals subjected to *rab-7* RNAi from hatching (whole-life RNAi)

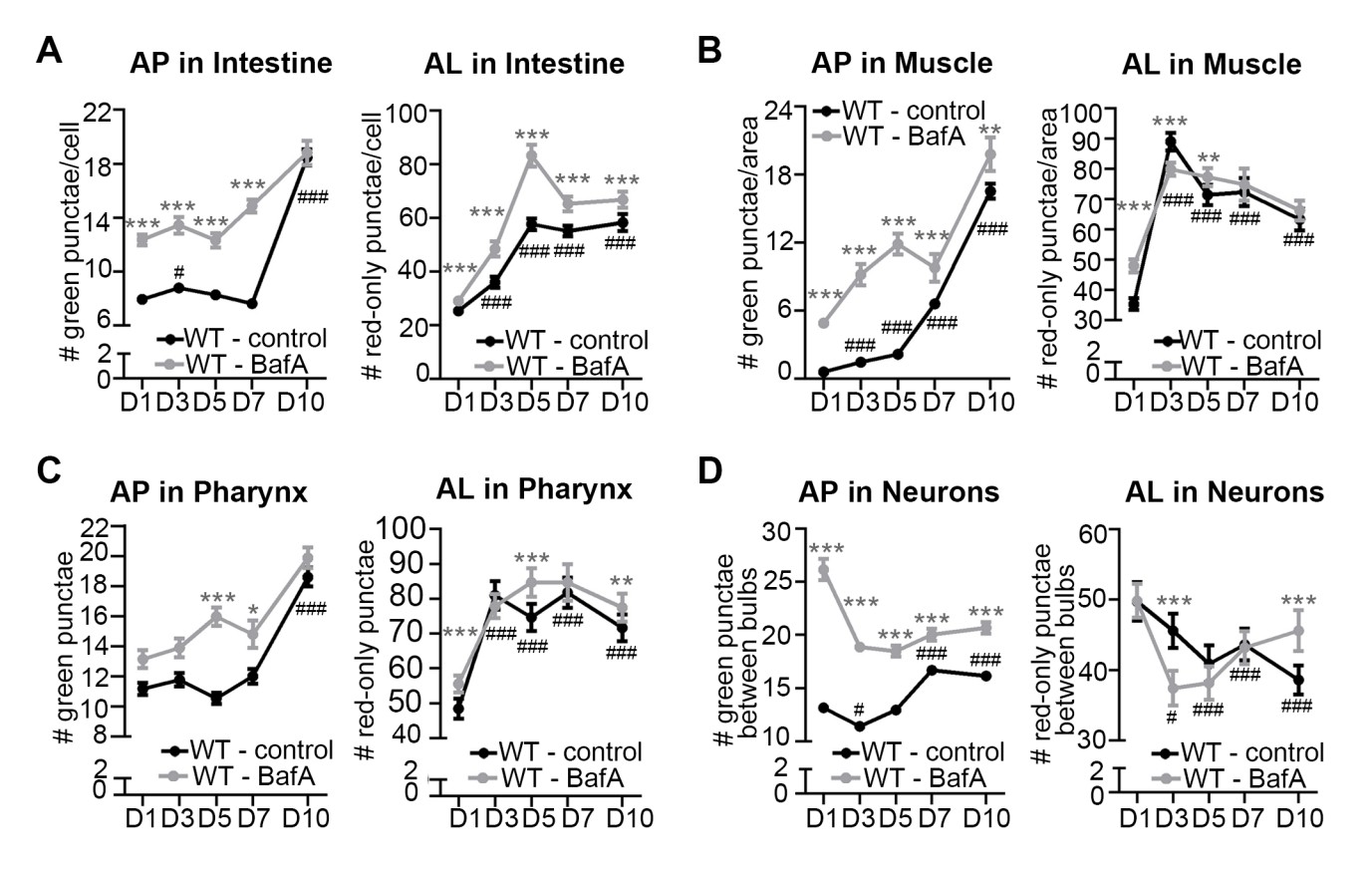

**Figure 3.** The pool size of autophagic vesicles increases with age in all tissues of wild-type animals. (A–D) Quantification of autophagosomes (AP) and autolysosomes (AL) in adult Days 1, 3, 5, 7, and 10 wild-type (WT) transgenic animals expressing *mCherry::gfp::lgg-1* and injected with DMSO (control, black lines) or Bafilomycin A (BafA, gray lines). Tissues examined were the intestine (A), body-wall muscle (B), pharynx (C), and nerve-ring neurons (D). WT animals were raised at 25°C and incubated at 20°C from Day 1 of adulthood. Data are the mean ± SEM of ≥30 animals combined from three independent experiments per time point. ***p<0.0001, **p<0.001, and *p<0.01 for WT control vs WT + BafA on each day; ###p<0.0001 for WT control at Days 1, 3, 5, 7, or 10 vs. WT control at Day 1 by Poisson regression.

The following figure supplements are available for figure 3:

**Figure supplement 1.** Quantification of autophagosomes at different time points after Bafilomycin A treatment and at different temperatures.

**Figure supplement 2.** Hypothetical outcomes of autophagosome and autolysosome pool sizes following Bafilomycin A treatment.

**Figure supplement 3.** Quantification of GFP::LGG-1 punctae during aging of animals expressing *gfp::lgg-1*.

contained increased numbers of APs and reduced ALs in their intestines, as would be expected when autophagy is active and AP–lysosome fusion is inhibited (*Figure 2K*). We also pharmacologically inhibited autophagy by injecting WT *mcherry::gfp::lgg-1* transgenic animals with Bafilomycin A (BafA), which blocks vacuolar type H⁺-ATPases thus inhibiting lysosomal acidification (*Figure 2J* and [*Klionsky et al., 2008*]). We confirmed this by showing that BafA abrogates LysoTracker staining in the intestine of Day 1 animals (*Figure 2—figure supplement 4A–B*). Consistent with a block in autophagy, we observed an increase in APs and a reduction in ALs in hypodermal seam cells of Day 1 transgenic WT animals injected with BafA compared with those injected with DMSO (*Figure 2L*). Lastly, we examined animals with a loss-of-function mutation in *cst-1*, the *C. elegans* ortholog of mammalian STK4; such mutants experience a block in autophagy (*Wilkinson et al., 2015*). While the number of APs in hypodermal seam cells of untreated mCherry::GFP::LGG-1-expressing *cst-1(tm1900)* mutants was higher than that of WT transgenic animals, as previously reported

(*Wilkinson et al., 2015*), we observed no change in AP or AL pool sizes following BafA treatment of these animals (*Figure 2—figure supplement 4C*), consistent with the pre-existing genetic block in autophagy. Collectively, these results support the notion that the new mCherry::GFP::LGG-1 reporter identifies both AP and AL compartments and can be used to monitor changes in these compartments upon modulation of autophagy in adult *C. elegans*.

## Aging of WT *C. elegans* is associated with an increase in the number of autophagic vesicles, which corresponds to a decrease in autophagic activity

We next characterized tissue-specific changes in AP and AL pool sizes in the intestine, muscle, pharynx, and neurons of adult Days 1, 3, 5, 7, and 10 WT animals expressing *mCherry::gfp::lgg-1*. Similar to our analysis of WT animals expressing *gfp::lgg-1* (*Figure 1F–I*), we observed that WT animals expressing mCherry::GFP::LGG-1 displayed an increase in AP numbers in the intestine, muscle, and pharynx during late adulthood (*Figure 3A–C*), with the largest (~15-fold) increase occurring in the muscle. Somewhat differently, the number of ALs in these tissues increased early in adulthood and then remained relatively constant until Day 10 (*Figure 3A–C*). In contrast, while neurons overall showed an increased number of APs when assessed with either the GFP::LGG-1 (*Figure 1I*), or the mCherry::GFP::LGG-1 reporter (*Figure 3D*), the number of ALs decreased with age (*Figure 3D*). Together, data with both fluorescent-tagged LGG-1 reporters indicate that the steady-state AP pool size increased with age in all the examined tissues of WT *C. elegans*, albeit with different trajectories, overall increasing confidence that APs are reliably monitored with the new tandem LGG-1 reporter. Likewise, the AL pool size appeared first to increase but then stagnate in the intestine, muscle, and pharynx of WT animals, whereas the number of ALs instead decreased over time in neurons.

An increase in APs and ALs could be due to an induction or a block in autophagy (*Klionsky et al., 2016*). To distinguish between these possibilities, we injected BafA into WT transgenic animals during adulthood and quantified APs and ALs in each tissue two hours later, when steady-state conditions were reached (*Figure 3—figure supplement 1A–D*). Since BafA inhibits lysosomal acidification (*Klionsky et al., 2008*), no change in AP and AL numbers following BafA treatment would be indicative of a block in autophagy, whereas BafA-induced changes in either AP or AL number, or both, would indicate active autophagy (*Figure 3—figure supplement 2*). Specifically, complete inhibition of lysosomal acidification by BafA would reduce AL numbers (mCherry-only punctae) while increasing AP numbers (mCherry/GFP punctae) since the GFP signal would no longer be quenched (*Figure 3—figure supplement 2Ai,B*). However, BafA may also prevent the later step of cargo degradation by incompletely inhibiting acidification of the lysosome resulting in reduced efficiency of lysosomal enzymes, which could lead to a concomitant increase in AL numbers (*Figure 3—figure supplement 2Aii,B* and *Ni et al., 2011*]). We found that BafA treatment increased AP numbers in all tissues and at all ages of *mCherry::gfp::lgg-1* transgenic animals compared with controls, but the increases were generally dampened in older animals (*Figure 3A–D*). Similar observations were made in BafA-injected *gfp::lgg-1* transgenic animals (*Figure 3—figure supplement 3A–H*; analyzed at Day 1 and Day 7). This dampening of the response was not due to BafA being less effective at older age, since BafA equally quenched Lysotracker in the intestine of Day 1 and Day 10 animals (*Figure 2—figure supplement 4A–B*, and data not shown). Interestingly, AL numbers were increased by BafA in the intestine and pharynx at all ages and in the muscle of young animals (*Figure 3A–C*), whereas the effect on neurons varied with time (*Figure 3D*). Collectively, the observed changes in AP and AL pool sizes following BafA treatment suggest that autophagy is active in all tissues of WT *C. elegans* throughout life, similar to conclusions from a recent study (*Chapin et al., 2015*). However, the observation that older animals generally showed reduced responses to BafA is consistent with an overall reduction in autophagic activity in all tissues with age. Although we cannot at this point explain the variable age-related effects of BafA on AL pool size in neurons, we note that the BafA response in both AP and AL compartments at Day 10 was larger in neurons than in other tissues, suggesting that autophagic capacity in neurons may decline less with age. This possibility remains to be directly tested.

Taken together, our analysis is consistent with the GFP::LGG-1 and mCherry::GFP::LGG-1 reporters similarly monitoring APs, that the AP and AL pool sizes generally increase with age, and that autophagy is active in all tissues throughout adulthood in WT *C. elegans*, consistent with

(*Chapin et al., 2015*). Importantly, however, the increased pool sizes appear to reflect an age-dependent decline in autophagic activity in all tissues.

## Autophagic activity in hypodermal seam cells is increased in *daf-2* mutants, but can not be accurately assessed in *glp-1* mutants

Mutations in the *daf-2* insulin/IGF-1 receptor or the *glp-1*/Notch receptor extend the lifespan of *C. elegans* (*Arantes-Oliveira et al., 2002*; *Kenyon et al., 1993*), and reported evidence suggests that this is dependent on induction of autophagy. However, it is not known whether *daf-2* and *glp-1* mutants differentially regulate autophagy in a tissue- or age-specific manner to secure lifespan extension. To begin to investigate this, we assessed autophagy in a spatiotemporal manner in long-lived *daf-2(e1370)* and *glp-1(e2141)* mutants expressing the different fluorescently-tagged LGG-1 reporters.

We first analyzed hypodermal seam cells, the cell type that, thus far, has been the most frequently monitored using the GFP::LGG-1 reporter in *C. elegans* (reviewed in *Zhang et al., 2015*; *Hansen, 2016*). These cells display an increased number of GFP::LGG-1-positive punctae in both *daf-2(e1370)* and *glp-1(e2141)* larvae (*Meléndez et al., 2003*; *Hansen et al., 2008*; *Lapierre et al., 2011*). Consistently, we found that Day 1 *daf-2(e1370)* or *glp-1(e2141)* mutants expressing either GFP::LGG-1 or mCherry::GFP::LGG-1 displayed an increased AP pool size compared with WT, whereas AL numbers were similar between WT and the long-lived mutants (*Figure 4—figure supplement 1A–G*). However, expression of lipidation-deficient GFP::LGG-1(G116A) unexpectedly resulted in GFP-positive punctae in hypodermal seam cells of *glp-1(e2141)* mutants, whereas WT and *daf-2(e1370)* animals, as expected, showed diffuse GFP::LGG-1(G116A) localization (*Figure 4—figure supplement 1D–E*). Notably, both BafA (*Figure 4—figure supplement 1A–C*) and *rab-7* RNAi (*Figure 4—figure supplement 1F*) treatments caused an increase in APs and a reduction in ALs in *daf-2(e1370)* mutants expressing mCherry::GFP::LGG-1, but had no effect on AP or AL numbers in *glp-1(e2141)* mutants (*Figure 4—figure supplement 1A–C,G*; we note that BafA was equally effective in quenching Lysotracker in the intestine of WT, *daf-2,* and *glp-1* animals at Day 1 and Day 10 (*Figure 2—figure supplement 4A–B,D*, and data not shown). Taken together, these data are consistent with autophagy being induced in hypodermal seam cells of *daf-2* mutants, as proposed previously (*Meléndez et al., 2003*). In contrast, hypodermal seam cells of *glp-1* animals may be different and cause LGG-1 to aggregate, as observed in autophagy-deficient mammalian cells expressing lipidation-deficient Atg8/LC3 (*Kuma et al., 2007*). Consistently, we observed punctate structures in *atg-3(bp412)* and *atg-18(gk378)* autophagy mutants expressing GFP::LGG-1(G116A) (*Figure 4—figure supplement 1H–I*). The exact nature of such lipidation-deficient GFP::LGG-1(G116A) punctate structures in *C. elegans* remains to be determined.

## *daf-2* mutants generally display increased autophagic activity

Since the GFP::LGG-1(G116A) reporter remained largely diffuse over time in the other major tissues we analyzed in *daf-2* and *glp-1* mutants, similar to WT animals (*Figure 1—figure supplement 1*), we next analyzed the GFP::LGG-1 and mCherry::GFP::LGG-1 reporters in the intestine, muscle, pharynx, and in neurons over time. Using both reporters, we observed the number of APs in the intestine and muscle were higher in *daf-2(e1370)* mutants than in WT animals at Day 1 and increased modestly with age until Day 10, at which time the AP levels in *daf-2(e1370)* mutants were lower than in WT animals (*Figure 4A–B*; *Figure 3—figure supplement 3A–B,E–F*). In contrast, while AL numbers were also elevated in the intestine and muscle of Day 1 *daf-2(e1370)* mutants with age (*Figure 4A–B*), AL numbers in the muscle increased in early adulthood and remained at relatively constant levels thereafter, whereas in the intestine, they continued to increase for longer. In both tissues, AL numbers in *daf-2(e1370)* mutants were higher than in WT animals at older ages (*Figure 4A–B*). Notably, BafA treatment generally increased the AP pool size in the intestine and muscle of *daf-2(e1370)* mutants expressing mCherry::GFP::LGG-1 (*Figure 4A–B*), or GFP::LGG-1 (*Figure 3—figure supplement 3E–F*). Of particular note, Day 10 animals showed the largest induction following BafA treatment in the intestine and muscle of *daf-2(e1370)* mutants, and AL pool size was also generally increased (*Figure 4A–B*). Collectively, these results suggest that autophagic activity may be elevated in the intestine and muscle of *daf-2* mutants compared with WT animals and may remain so throughout adulthood.

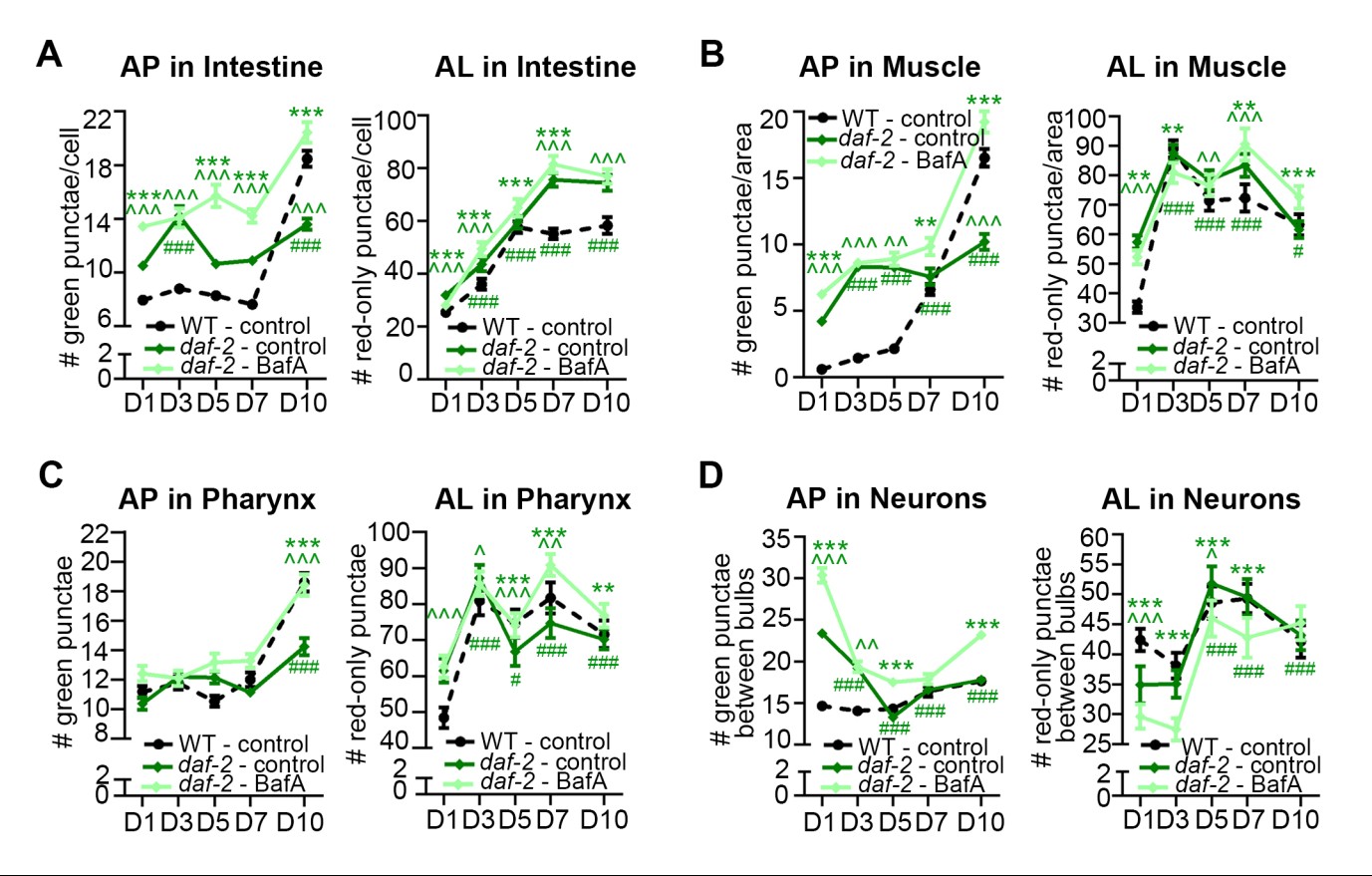

**Figure 4.** *daf-2* insulin/IGF-1 receptor mutants display increased autophagic activity in most tissues. (**A–D**) Quantification of autophagosomes (AP) and autolysosomes (AL) in adult Day 1, 3, 5, 7, and 10 *daf-2(e1370)* animals expressing *mCherry::gfp::lgg-1* and injected with DMSO (control, dark green lines) or Bafilomycin A (BafA, light green lines). Tissues examined were the intestine (**A**), body-wall muscle (**B**), pharynx (**C**), and nerve-ring neurons (**D**). The black dashed lines in (**A–C**) show data from wild-type (WT) control animals from *Figure 3* for comparison (animals were analyzed in parallel). The black dashed line in (**D**) shows data from WT animals incubated at 20℃ for their entire lifespan. Data are the mean ± SEM of ≥25 animals combined from three independent experiments. ^, WT control vs. *daf-2* control at Days 1, 3, 5, 7, and 10; *, *daf-2* control vs. *daf-2* + BafA at Days 1, 3, 5, 7, and 10; #, *daf-2* control at Days 3, 5, 7, and 10 vs. *daf-2* control at Day 1. ***/^^^/###p<0.0001, **/^^/##p<0.001, */^/#p<0.01 by Poisson regression. See also *Figure 3—figure supplement 1A–H* for quantification of APs in *gfp::lgg-1* transgenic animals.

The following figure supplement is available for figure 4:

**Figure supplement 1.** Hypodermal seam cells in *daf-2* mutants display increased autophagy, whereas lipidation-independent punctate structures are present in these cells in *glp-1* mutants.

In the pharynx, the AP pool size was similar in *daf-2(e1370)* and WT animals at most time points, whereas the AL pool size was initially larger in *daf-2(e1370)* mutants; however, in older *daf-2(e1370)* mutants, both AP and AL numbers were similar or lower than WT animals (*Figure 4C*, *Figure 3—figure supplement 3C,G*). Interestingly, BafA treatment did not cause any significant changes in AP numbers in the pharynx of *daf-2(e1370)* mutants, except at Day 10 of adulthood (*Figure 4C*, *Figure 3—figure supplement 3C,G*), whereas AL numbers were changed from Day 5 to Day 10 (*Figure 4C*). Collectively, these data indicate the interesting possibility that autophagy in the pharynx of *daf-2* mutants may be blocked prior to Day 5 of adulthood. Moreover, while enumeration of APs alone would suggest a continuous block in autophagy in the pharynx of *daf-2* mutants, the AL flux data is consistent with possible reactivation of autophagic activity that exceeds that of WT in late life. Additional experiments are needed to evaluate this possibility and to conclusively interpret the AL flux data, especially at older age.

The effect of age on the GFP::LGG-1 and mCherry::GFP::LGG-1 reporters in neurons was quite different to that in other tissues in *daf-2* mutants, but similar to that observed in WT animals. APs and ALs were more and less abundant, respectively, in *daf-2(e1370)* animals than in WT animals in early adulthood, but both strains displayed an overall decrease in APs and an increase in ALs as they aged (*Figure 4D*, *Figure 3—figure supplement 3D,H*). BafA treatment altered both AP and AL numbers throughout early adulthood in *daf-2(e1370)* animals (*Figure 4D*, *Figure 3—figure supplement 3H*), while at Day 10, BafA altered APs but not ALs (*Figure 4D*). Collectively, these data are consistent with autophagic activity in the neurons of *daf-2* mutants possibly being higher than in WT animals in early adulthood, whereas both strains display an age-related impairment of autophagic activity over time.

Overall, these data indicate that autophagic activity in the intestine, muscle, and neurons may be higher in *daf-2* mutants than in WT animals, at least in early adulthood. Notably, activity in the pharynx may possibly be blocked in younger *daf-2* mutants. This observation constitutes the first report of a potential block in autophagy in a long-lived animal, and needs to be investigated in more detail.

## *glp-1* animals display a different autophagic activity profile than *daf-2* mutants

We performed a similar spatiotemporal analysis of the autophagy reporters in the major tissues of long-lived, germline-less *glp-1* mutants. We previously reported that GFP::LGG-1 punctae are more abundant in the intestine of *glp-1* mutants than of WT animals at Day 1 of adulthood (*Lapierre et al., 2011*). Consistent with this, we observed elevated numbers of APs in the intestine of *glp-1(e2141)* animals compared with WT animals expressing either GFP::LGG-1 or mCherry::GFP:: LGG-1 throughout adulthood except at old age (*Figure 5A*, *Figure 3—figure supplement 3A,E'*). In contrast, the intestinal AL pool size was lower in *glp-1(e2141)* mutants than in WT animals at Day 1 of adulthood and fluctuated thereafter (*Figure 5A*). BafA treatment had variable effects on AP numbers but increased AL numbers in the intestine of both young and old animals (*Figure 5A*, *Figure 3—figure supplement 3E'*). These results are consistent with autophagy being active in the intestine of *glp-1* animals throughout life, although this dataset is insufficient to make a direct comparison of the relative autophagic activity in *glp-1* mutants and WT animals.

In muscle, AP numbers were higher in *glp-1(e2141)* mutants than in WT animals in early adulthood but they increased similarly in both strains thereafter (*Figure 5B*, *Figure 3—figure supplement 3B, F'*). In contrast, AL numbers were generally lower in *glp-1(e2141)* mutants than in WT animals and remained relatively constant over time (*Figure 5B*). In this tissue, the BafA treatment, for unknown reasons, had variable effects on the two different reporter strains. Whereas *glp-1(e2141)* animals expressing GFP::LGG-1 showed induction of APs at both Day 1 and Day 7 (*Figure 3—figure supplement 3F'*), BafA had no significant effect on APs at any time point in *glp-1(e2141)* mutants expressing mCherry::GFP::LGG-1 (*Figure 5B*). ALs were similarly unaffected by BafA treatment in Days 1–5 *glp-1(e2141)* animals, yet altered in Day 7 and Day 10 *glp-1(e2141)* animals (*Figure 5B*). Thus, while additional experiments are needed to conclusively evaluate autophagy activity in the muscle of younger *glp-1* mutants, our observations indicate that autophagy may be active in this tissue of older *glp-1* mutants.

In the pharynx, *glp-1(e2141)* and WT animal's AP numbers were generally similar throughout adulthood (*Figure 5C*, *Figure 3—figure supplement 3C,G'*), whereas the number of ALs increased somewhat with age but was consistently lower than in WT animals after Day 1 (*Figure 5C*). AP numbers in *glp-1(e2141)* mutants were generally unaffected by BafA treatment, whereas AL counts were altered in animals irrespective of age (*Figure 5C*, *Figure 3—figure supplement 3C,G'*). Thus, while the AL compartment needs to be fully investigated in older animals, it is possible that autophagy may remain active in the pharynx of *glp-1* mutants throughout life. However, similar to the intestine, it is not possible to evaluate the relative autophagic activity in *glp-1* mutants to WT animals based on this dataset.

In neurons, APs were more abundant in *glp-1(e2141)* mutants than in WT animals at Day 1, but were similar in older animals (*Figure 5D*, *Figure 3—figure supplement 3D,H'*). ALs were less abundant in the neurons of young Day 1 and Day 3 *glp-1(e2141)* mutants compared with WT animals, but ALs gradually decreased with age in WT animals while they remained relatively constant in *glp-1(e2141)* mutants (*Figure 5D*). Of note, BafA treatment of young *glp-1(e2141)* animals expressing either of the two LGG-1 reporters altered both AP and AL pool sizes (*Figure 5D*,

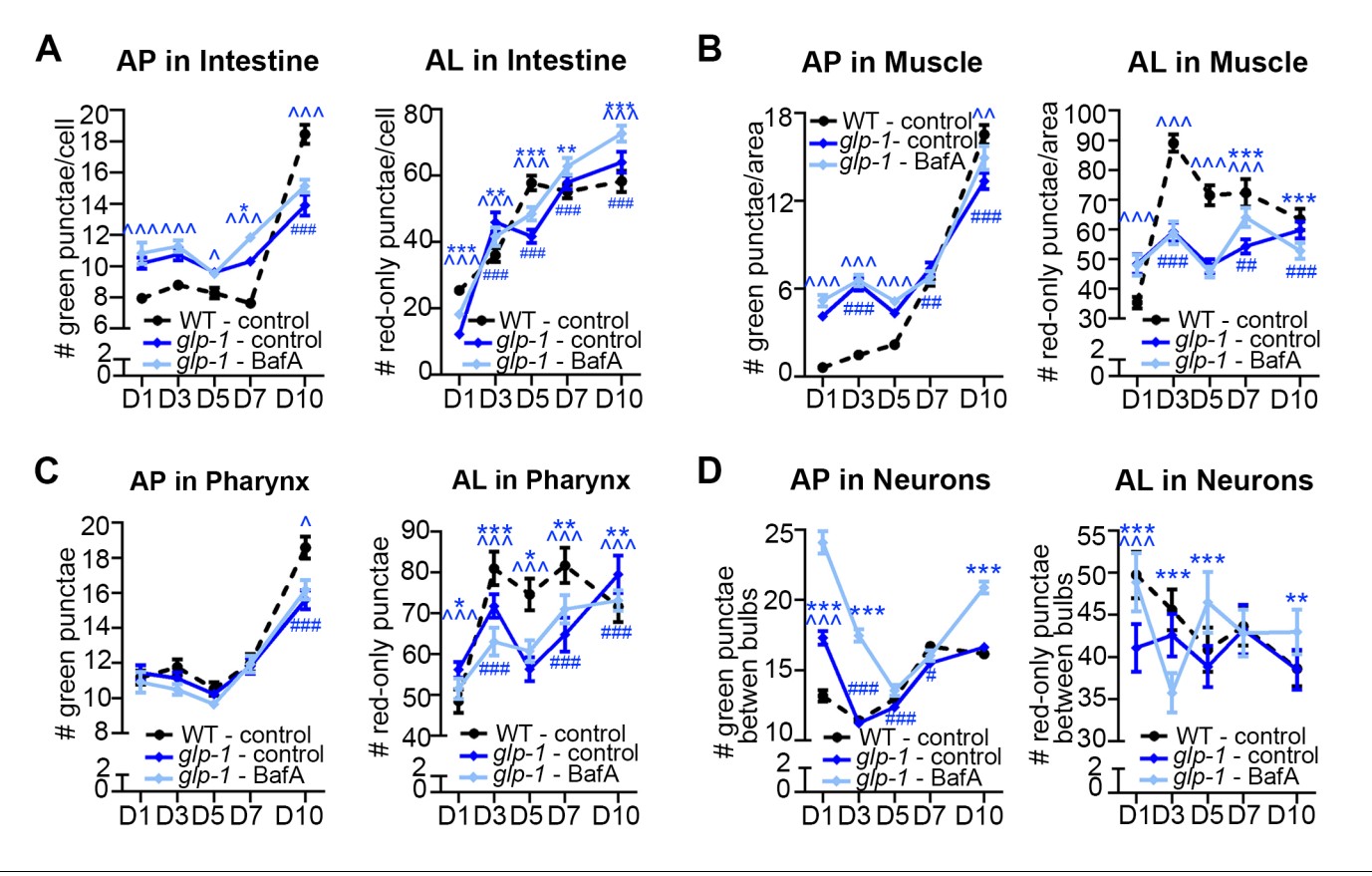

**Figure 5.** Germline-less *glp-1* mutants display a different autophagic activity profile than *daf-2* mutants. (**A–D**) Quantification of autophagosomes (AP) and autolysosomes (AL) in adult Days 1, 3, 5, 7, and 10 *glp-1(e2141)* animals expressing *mCherry::gfp::lgg-1* and injected with DMSO (control, dark blue lines) or Bafliomycin A (BafA, light blue lines). Tissues examined were the intestine (**A**), body-wall muscle (**B**), pharynx (**C**), and nerve-ring neurons (**D**). The black dashed lines in (**A–D**) show data from wild-type (WT) control animals from *Figure 3* for comparison (animals were analyzed in parallel). Data are the mean ± SEM of ≥25 animals combined from three independent experiments. ^, WT + control vs. *glp-1* control at Days 1, 3, 5, 7, and 10; *, *glp-1* control vs. *glp-1* + BafA at Days 1, 3, 5, 7, and 10, #, *glp-1* control at Days 3, 5, 7, and 10 vs. *glp-1* control at Day 1. ***/^^^/###p<0.0001, **/^^/##p<0.001, */^/#p<0.01 by Poisson regression analysis.

*Figure 3—figure supplement 3H'*), caused no significant effect on AP or AL number on Day 7 (*Figure 5D*, *Figure 3—figure supplement 3H'*), but resulted in a significant increase in AP and ALs at Day 10 (*Figure 5D*). Taken together, these data indicate that autophagic activity may be increased in the neurons of young *glp-1* animals compared to age-matched WT animals, whereas the autophagic activity may become more comparable to that in WT animals by mid-life.

Collectively, these data suggest that autophagy remains active in the intestine, pharynx, and possibly also muscle of *glp-1* mutants throughout adulthood. In neurons, autophagy appeared to be increased in young *glp-1* animals compared to WT animals and remains active and potentially comparable to WT by mid-life. Overall, aging had different tissue-specific effects of autophagy in *glp-1* mutants versus *daf-2* mutants, highlighting the possibility, which needs to be further addressed by yet-to-be developed biochemical assays, that autophagy may be regulated differentially at the individual tissue level in long-lived mutants.

## Autophagy gene expression in the intestine is required for lifespan extension of *glp-1* mutants but not of *daf-2* mutants

The intestine is a critical tissue in both *daf-2* and *glp-1* mutants, since the downstream effector DAF-16, a FOXO transcription factor functions in this tissue to ensure lifespan extension (*Libina et al., 2003*). Since our tissue analyses indicated that autophagy is active in the intestine

of both mutants, we asked whether intestine-specific inhibition of autophagy prevents lifespan extension in *daf-2* and *glp-1* animals, as has been observed with whole-body inhibition of autophagy (*Hars et al., 2007*; *Lapierre et al., 2011*; *Meléndez et al., 2003*; *Hansen et al., 2008*). To this end, we used *sid-1(qt9)* mutants, which carry a mutation in the dsRNA transporter *sid-1* rendering them systemically refractory to RNAi, but with reconstituted *sid-1* expression specifically in the intestine using the *vha-6* promoter (*Melo and Ruvkun, 2012*). This strain was crossed to *daf-2(e1370)* and *glp-1(e2141)* mutants and lifespan analyses were carried out. The lifespans of the RNAi-refractory strains *daf-2(e1370); sid-1(qt9)* and *glp-1(e2141); sid-1(qt9)* with or without intestinal-specific *sid-1* re-expression were similar to the lifespans of *daf-2(e1370)* and *glp-1(e2141)* single mutants, respectively (*Supplementary file 1A–B*). Moreover, the lifespan of *daf-2(e1370); sid-1(qt9)* and *glp-1(e2141); sid-1(qt9)* double mutants were not affected by *atg-18/Wipi* RNAi (*Supplementary file 1A–B*), whereas the double mutants expressing *vha-6p::sid-1* were RNAi competent in the intestine, but not in the other tissues examined (*Figure 6—figure supplement 1A–E*, and data not shown). Together, these results validate the use of these strains for tissue-specific lifespan analyses.

While whole-body and intestinal-specific RNAi of *daf-16/Foxo* potently shortened the lifespan of both *glp-1(e2141)* and *daf-2(e1370)* mutants (*Figure 6A–B*; *Supplementary file 1A-B*), consistent with *daf-16* functioning in the intestine to extend the lifespan of these long-lived animals (*Libina et al., 2003*), we found striking differences in longevity of *glp-1* and *daf-2* animals subjected to intestinal *atg-18* RNAi. Specifically, we observed that intestinal-specific inhibition of autophagy by *atg-18/Wipi* RNAi was sufficient to significantly reduce the lifespan of *glp-1(e2141)* animals, as previously observed with whole-body *atg-18/Wipi* RNAi (*Figure 6C*, *Supplementary file 1A*, and *Lapierre et al., 2011*). In contrast, knockdown of *atg-18/Wipi* or of *lgg-1/Atg8* in the intestine of *daf-2(e1370)* animals did not have a significant effect on lifespan in a total of six out of six experiments, whereas whole-body *atg-18/Wipi* and *lgg-1/Atg8* RNAi had significant lifespan-shortening effects (*Figure 6D*, *Supplementary file 1B*), similar to the effect of whole-body RNAi of other autophagy genes (*Meléndez et al., 2003*; *Hansen et al., 2008*; *Hars et al., 2007*). Notably, we confirmed that *atg-18/Wipi* RNAi reduced the expression of an *mCherry::atg-18* transgene to the same extent in the intestine of WT and in *daf-2(e1370)* mutants (*Figure 6—figure supplement 1F,G*), indicating that knockdown of *atg-18* by RNAi is not somehow compromised in the intestine of *daf-2* mutants. Collectively, these data therefore indicate that intestinal autophagy is required for the lifespan extension of *glp-1* mutants, but not *daf-2* mutants.

This was a surprising result, because our autophagy flux assays indicated that *daf-2* mutants displayed active autophagy in the intestine (*Figure 4A*, *Figure 3—figure supplement 3A,E*), similar to *glp-1* mutants (*Figure 5A*, *Figure 3—figure supplement 3A,E'*). Autophagy in other tissues may instead play a role in the longevity of *daf-2* mutants. In support of this, we found that muscle-specific knockdown of *atg-18/Wipi* decreased the lifespan of *daf-2(e1370); sid-1(qt9)* animals expressing *sid-1* from the *myo-3* promoter (*Supplementary file 1B*). Of note, we observed the same lifespan-shortening effect of muscle-specific *atg-18/Wipi* RNAi in *glp-1(e2141)* mutants; in this case, however, *glp-1(e2141); sid-1(qt9)* mutants with *sid-1* re-expression in muscle were unexpectedly shorter-lived than *glp-1(e2141); sid-1(qt9)* mutants (*Supplementary file 1A*). Since the reason for this remains unclear, we refrain from drawing conclusions from this observation. Taken together, our lifespan analyses suggest that intestinal autophagy is required for lifespan extension in *glp-1* mutants, whereas autophagy in the muscle, but not the intestine, may contribute to the longevity of *daf-2* mutants.

## Discussion

Changes in autophagy have been linked to aging in many species, but it is not yet clear how autophagy is modulated spatially and temporally during an animal's lifespan. Our analysis of *C. elegans* using multiple Atg8 reporters and autophagy flux assays is the first effort to comprehensively estimate autophagic activity in a living animal during aging. We found that (i) aging of wild-type (WT) animals is accompanied by increased numbers of autophagic vesicles in all tissues examined that likely reflects a reduction in autophagy activity; (ii) long-lived *daf-2* insulin/IGF-1 mutants and germline-less *glp-1* mutants differentially regulate autophagy spatially and temporally compared to WT animals, and (iii) *glp-1* mutants, but not *daf-2* mutants, require autophagy genes in the intestine for

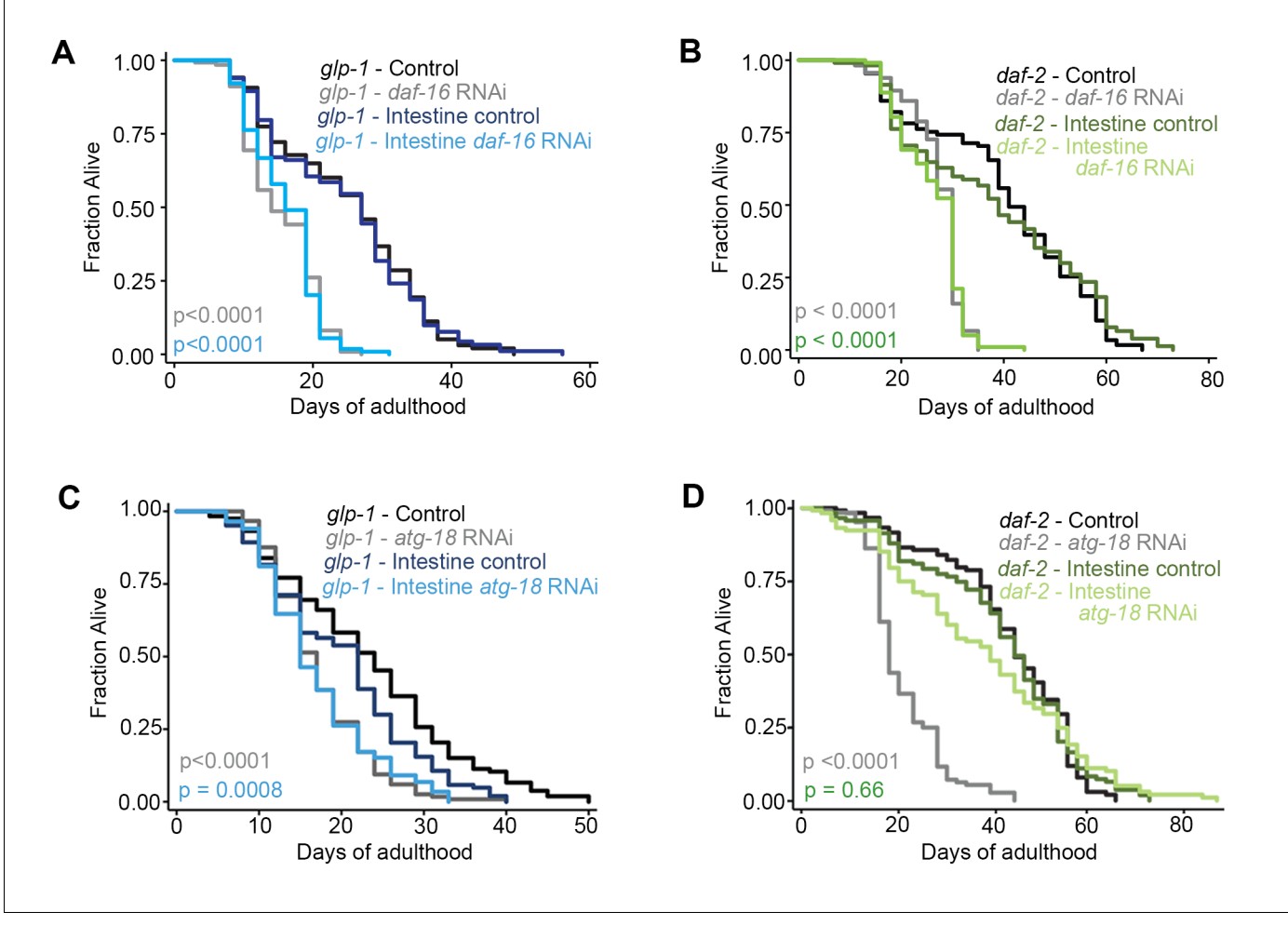

**Figure 6.** Autophagy genes expressed in intestinal cells are required for lifespan extension of *glp-1* mutants, but not of *daf-2* mutants. (**A,C**) Kaplan–Meier survival curves of *glp-1(e2141)* single mutants and *glp-1(e2141); sid-1(qt9)* double mutants expressing *sid-1* cDNA in the intestine (*vha-6* promoter). Animals were fed from Day 1 of adulthood with bacteria expressing empty vector (control), *daf-16/Foxo* dsRNA (**A**), or *atg-18/Wipi* dsRNA (**C**). Intestine-specific inhibition of *daf-16/Foxo* shortened the lifespan of *glp-1(e2141)* mutants in two out of two experiments ach with ≥100 animals. p<0.0001 for whole body control RNAi vs. whole body *daf-16/Foxo* RNAi; p<0.0001 for intestine-specific control RNAi vs. intestine-specific *daf-16* RNAi by log-rank test. Intestine-specific inhibition of *atg-18/Wipi* shortened the lifespan of *glp-1(e2141)* mutants in five out of seven experiments, each with ≥100 animals. p<0.0001 for whole-body control RNAi vs. whole-body *atg-18/Wipi* RNAi; p=0.0008 for intestine-specific control RNAi vs. intestine-specific *atg-18/Wipi* RNAi by log-rank test. (**B,D**) Kaplan–Meier survival curves of *daf-2(e1370)* single mutants and *daf-2(e1370); sid-1(qt9)* double mutants expressing *sid-1* cDNA in the intestine (*vha-6* promoter). Animals were fed from Day 1 of adulthood with bacteria expressing empty vector (control), *daf-16/Foxo* dsRNA (**B**), or *atg-18/Wipi* dsRNA (**D**). Intestine-specific inhibition of *daf-16/Foxo* shortened the lifespan of *daf-2(e1370)* mutants in two out of two experiments, each with ≥100 animals. p<0.0001 for whole body control RNAi vs. whole body *daf-16/Foxo* RNAi; p<0.0001 for intestine-specific control RNAi vs. intestine-specific *daf-16* RNAi by log-rank test. Intestine-specific inhibition of *atg-18/Wipi* had no significant effect on the lifespan of *daf-2(e1370)* mutants in all of six experiments, each with ≥100 animals. p<0.0001 for whole-body control RNAi vs. whole-body *atg-18/Wipi* RNAi; p=0.66 for intestine-specific control RNAi vs. intestine-specific *atg-18/Wipi* RNAi, by log-rank test. See ***Supplementary file 1*** for details on lifespan analyses and additional repeats.

The following figure supplement is available for figure 6:

**Figure supplement 1.** Characterization of intestinal RNAi strains.

lifespan extension, further emphasizing important differences in autophagy regulation in individual tissues between these conserved longevity paradigms.

For this study, we used previously published GFP::LGG-1 (***Gelino et al., 2016***; ***Kang et al., 2007***; ***Meléndez et al., 2003***) and new mCherry::GFP::LGG-1 reporters (which were expressed from

endogenous or neuronal promoters) to quantify autophagic events in four major somatic tissues, that is, intestine, muscle, pharynx, and neurons during *C. elegans* adulthood. As an important control to help verify that GFP::LGG-1-positive punctae observed in WT, *daf-2*, and *glp-1* animals likely represented APs and not unspecific aggregates, we also used a lipidation-deficient GFP::LGG-1(G116A) reporter expressed from the endogenous promoter (*Manil-Ségalen et al., 2014*). Similar point-mutated Atg8 proteins are assumed unable to attach to the autophagosomal membrane since they cannot be post-translationally modified to bind phosphatidylethanolamine (*Kabeya et al., 2004*). Consistent with this, GFP::LGG-1(G116A) appeared diffusely localized in essentially all the settings we investigated here, that is, in the intestine, muscle, and pharynx, with the notable exception of hypodermal seam cells in *glp-1* mutants in which GFP::LGG-1(G116A) expression led to punctae formation, similar to *glp-1* mutants expressing GFP::LGG-1 (*Lapierre et al., 2011*). GFP::LGG-1 punctae in hypodermal seam cells of *glp-1* mutants may therefore have been incorrectly interpreted as APs. While GFP::LGG-1(G116A) punctate structures in *C. elegans* remains to be investigated in detail, they may be similar in nature to the intracellular protein aggregates or inclusion bodies observed in mammalian Atg5 knockout cells expressing GFP-tagged Atg8/LC3 (*Kuma et al., 2007*). Since Atg8 proteins are capable of oligomerization (*Nymann-Andersen et al., 2002*; *Baisamy et al., 2009*), GFP::LGG-1(G116A) punctate structures may alternatively still represent APs with mutant LGG-1(G116A) protein. However, methods to concretely determine whether GFP::LGG-1(G116A) punctae represent aggregates or APs (with WT and mutant LGG-1 protein) are not currently available in *C. elegans*. Collectively, these observations emphasize the use of analyzing a LGG-1(G116A) reporter in any given condition in which a fluorescent LGG-1 reporter is utilized to help assess autophagic vesicles, while the data from these reporters should be interpreted with caution. To this end, we note that a comprehensive analysis of such a reporter in neurons needs to be carried out since the GFP::LGG-1(G116A) reporter we used here limited us to only evaluate hypodermal seam cells, intestine, muscle, and pharynx in detail.

We used the GFP::LGG-1 and mCherry::GFP::LGG-1 reporters in combination with Bafilomycin (BafA) flux assays to quantify AP and AL numbers and estimate autophagic activity. These reporters equivalently monitored changes in the AP compartment in all tissues and at all ages, collectively increasing confidence that the AP compartment was accurately visualized with these reporters. Moreover, the mCherry::GFP::LGG-1 reporter rescued an embryonically lethal *lgg-1/Atg8* mutant, indicating that the full-length protein is functional. We sought to use this new reporter to monitor the AL compartment for the first time in adult *C. elegans* after validating it in multiple ways Specifically, we found that co-staining with Lysotracker highlighted the majority of red-only punctae in the intestine of young WT animals, as expected for acidic autolysosomes. In addition, we observed that inhibition of autophagy by either BafA, which inhibits lysosomal acidification, RNAi of *rab-7*, a gene required for AP-lysosome fusion, or genetic inhibition of autophagy by mutation of the Hippo kinase *cst-1*, all modulated AP and AL numbers in young WT and long-lived mutants as expected. While these observations collectively support the use of mCherry::GFP::LGG-1 to reliably monitor the AP and AL compartments, we were unable to account for ~15% red-only punctae in the intestine of Day 1 WT animals expressing mCherry::GFP::LGG-1. A similar low percentage of red-only punctae was observed in the intestine of Day 1 *daf-2* and *glp-1* mutants (*Figure 2—figure supplement 4D*). These punctae could represent aggregates, or, alternatively, inefficient Lysotracker staining (as noted in results), however, at present we cannot distinguish between these possibilities and this estimate does not significantly offset any of our conclusions. While this estimate is very low, and has, to our knowledge, not previously been estimated for a tandem-tagged Atg8 reporter in any model system including mammalian cell culture, we emphasize that additional experiments are needed to fully verify the AL compartment in other *C. elegans* tissues and at later time points. To this end, it will be important to evaluate a lipidation-deficient mCherry::GFP::LGG-1(G116A) mutant. Moreover, it would be of interest to estimate autophagic activity via the Atg8 paralog LGG-2 in *C. elegans* (*Alberti et al., 2010*).

Our BafA flux assays indicated that although autophagy is active in the intestine, muscle, pharynx, and neurons of aging WT *C. elegans,* autophagic activity appeared to decrease with age. These results are consistent with flux assays and proteolysis experiments in rat liver (*Del Roso et al., 2003*, *Donati et al., 2001*), gene expression studies in *D. melanogaster* and rodents (*Cuervo and Wong, 2014*; *Demontis and Perrimon, 2010*; *Kaushik et al., 2012*; *Ling and Salvaterra, 2009*; *Simonsen et al., 2008*; *Vittorini et al., 1999*; *Ye et al., 2011*), and lysosomal protease studies in *C.*

*elegans* (*Sarkis et al., 1988*). Our observations are also consistent with the findings of a recent study of tissue-specific autophagy in *C. elegans*, which showed that autophagy is active in all major somatic tissues during aging (*Chapin et al., 2015*). In that study, *C. elegans* expressing a lysosomal-cleavable dual-fluorescent LGG-1 protein showed an age-associated increase in mono-fluorescent LGG-1 protein levels, which the authors interpreted as an increase in autophagic activity (*Chapin et al., 2015*). However, the accumulation of this mono-fluorescent protein would also be consistent with an age-dependent decline in autophagic activity if degradation is inhibited, perhaps through reduced lysosomal pH or enzymatic activity, as has been observed in aging yeast (*Hughes and Gottschling, 2012*). How lysosomal pH and activity might change in individual *C. elegans* tissues during aging is an important question for the future and will be especially interesting to investigate in neurons, which appeared to behave differently to other major tissues with respect to autophagic activity. The apparent decline in autophagic activity observed here could have several explanations, including an age-associated failure to clear the autophagic machinery (as evidenced by the increases in AP and AL pool sizes) or to degrade otherwise detrimental cargo. Additional assays are required to confirm these interpretations, including biochemical approaches to directly address how autophagy might fail with age in the different tissues of *C. elegans*; such efforts may be facilitated by novel tissue-isolation methods (*Kaletsky et al., 2016*). It will also be interesting to investigate how an age-dependent decline of autophagy in specific tissues might contribute to loss of organ-specific functions during aging.

Our analysis suggested that autophagic activity might be increased in certain tissues of long-lived *daf-2* and *glp-1* mutants compared with WT animals, as we and others have previously proposed (*Lapierre et al., 2011*; *Meléndez et al., 2003*). However, our results indicate that these two longevity mutants differentially regulate autophagy in a tissue-specific manner during aging since *daf-2* and *glp-1* mutants showed distinct age-related changes in AP and AL pool sizes, as well as in autophagic flux. Of particular note, we found that autophagy appeared to be blocked in the pharynx of Day 1 adult *daf-2* animals. This is an interesting scenario as it would constitute the first example of a possible autophagy block in a long-lived animal; however, we caution that steady-state conditions for the BafA treatment could vary between genetic background, tissue, as well as age. Irrespectively, it remains to be determined how *daf-2* and *glp-1* mutants may differentially regulate autophagy, and to this point, it is interesting to note that although both mutants require the transcription factor HLH-30/TFEB for longevity (*Lapierre et al., 2013*), *glp-1* mutants upregulate more predicted HLH-30 target genes with roles in autophagy than *daf-2* mutants (*Lapierre et al., 2013*). While the endpoint of our analysis was on Day 10 of adulthood, it will also be of interest to estimate autophagic activity in physiologically old long-lived mutants, including in their late state of decrepitude likely caused by bacterial colonization (*Podshivalova et al., 2017*).

We observed that intestine-specific inhibition of autophagy genes significantly shortened the lifespan of *glp-1* animals, but not *daf-2* animals, suggesting that the apparent increase in intestinal autophagic activity in *daf-2* mutants may not be required for their long lifespan. Since systemic inhibition of autophagy reduces the lifespan of both mutants (reviewed in *Hansen, 2016*, and this study), autophagy in other tissues, such as muscle, may be more important for longevity in *daf-2* mutants. These results further highlight important differences in autophagy regulation between *daf-2* and *glp-1* mutants and suggest that autophagy in different organs may have distinct biological roles; a possibility that is further emphasized by the observed differential requirement for autophagy in lifespan determination in the intestine of *daf-2* versus *glp-1* animals. To this end, it will be interesting to determine whether autophagy is regulated in a tissue-specific manner in other long-lived *C. elegans* mutants, or in other long-lived species.

In this study, we evaluated overall autophagic activity by quantifying steady-state levels of APs and ALs in combination with autophagic flux assays. However, autophagy is a multistep process and further insight can be gained into its regulation by taking into account that each step occurs at a distinct rate. Below, we reassess our dataset with this in mind. If we assume that autophagy can be represented by three steps: isolation membrane (IM) to AP, AP to AL, and degradation of ALs, the rates of each step under steady-state conditions can be defined as the concentration of IMs, APs, and ALs multiplied by the rate constants $\alpha$, $\beta$, and $\gamma$, respectively (*Figure 2—figure supplement 3A–B* and [*Loos et al., 2014*]). Changes in the steady-state levels of APs and ALs can therefore provide information about the relative contribution of the change in each step to the overall autophagic flux. For example, in WT *C. elegans* under steady-state conditions, ALs were consistently more abundant

than APs in all tissues examined, as has been observed in some mammalian cell types (*Klionsky et al., 2016*), and in *D. melanogaster* tissues (*Mauvezin et al., 2014*). Assuming α is constant, this observation suggests that β is greater than γ (i.e. the degradation step is slower than the AP to AL step), implying that turnover of ALs may be rate limiting in *C. elegans* (*Figure 2—figure supplement 3B–C*). We can similarly approximate how BafA treatment affects each step and rate constant since the ratio of [AP] in control-treated animals to [AP] in BafA-treated animals (and similarly for [AL]) is inversely correlated to the change in β (or γ). For example, an increase in both [AP] and [AL] following BafA treatment implies that BafA reduces both β and γ (*Figure 2—figure supplement 3B,D*). In fact, we observed that each tissue in WT, *daf-2*, and *glp-1* animals responded to BafA in a different manner according to the genetic background and age of the animal, suggesting that tissue-, age-, and genotype-specific differences may exist in the autophagy-rate constants. The above analysis also clearly indicates that, while quantification of autophagic vesicles at steady state provides valuable information about autophagic activity, it at times is not sufficient to accurately determine the relative difference in autophagic activity between WT, *daf-2*, and *glp-1* animals, or between the tissues examined. Moreover, a hypothetical modeling of our data shows that steady-state numbers of APs and ALs can be interpreted as increased, decreased, or unchanged autophagic flux when the rate constants differ (*Figure 2—figure supplement 3E*), again cautioning that steady-state measurements of AP or AL pool sizes may not reflect the overall autophagic flux. Therefore, additional methods that directly measure the rate of at least one step in autophagy or the overall rate of autophagic degradation will be crucial to accurately measure autophagic flux and fully understand its role in longevity.

In conclusion, our autophagic flux analysis using LGG-1/Atg8 reporters indicated an age-dependent decline in autophagy in several major tissues of *C. elegans*, and that the reduction occurs at a step after AP formation. In contrast, long-lived animals show differential regulation of autophagy in distinct tissues. Further experiments are needed to confirm these observations by directly measuring autophagic activity. Understanding the tissue- and age-specific regulation of autophagy in *C. elegans* is likely to shed light on the role of autophagy in aging and age-related diseases in mammals, including humans.

## Materials and methods

### *C. elegans* and bacterial strains

*C. elegans* strains were maintained and cultured under standard conditions at 20°C on the standard *E. coli* strain OP50 (see *Supplementary file 2* for strain list). HT115 was used as the food source for feeding RNAi experiments (*Brenner, 1974*). For experiments with the temperature-sensitive *glp-1(e2141)* mutant, both *glp-1(e2141)* and wild-type (WT) N2 animals were allowed to hatch at 20°C for 24 hr and then moved to 25°C until Day 1 of adulthood. At Day 1 of adulthood, all strains were placed at 20°C for the rest of the analysis. All *daf-2(e1370)* strains were maintained at 20°C for their entire lifespan. WT animals were raised at 25°C and placed at 20°C at Day 1 of adulthood, with the exception of WT animals expressing the mCherry::GFP::LGG-1 transgene specifically in neurons (*Figure 4D*) or otherwise noted, which were maintained at 20°C for their entire lifespan. No significant difference was observed in AP and AL pool sizes in Day 1 or Day 3 adult animals raised at 20°C or 25°C (*Figure 3—figure supplement 1E–H*).

### RNA interference (RNAi)

HT115 RNAi clones used were *bli-3*, *bli-4*, *elt-2*, *lin-26*, *lgg-1*, *pept-1*, *rab-7*, and *unc-112* (all Ahringer library), *atg-18* (Vidal library), as well as *daf-16* and *gfp* (kind gifts from Dr. Andrew Dillin). All RNAi bacterial clones were verified by sequencing. The empty vector (L4440) for controls was provided by Dr. Andrew Fire. RNAi experiments were carried out as previously described (*Gelino et al., 2016*). Briefly, HT115 bacteria were grown in liquid LB medium containing 0.1 mg/ml carbenicillin (BioPioneer, San Diego, CA), 80 μl bacteria spotted onto 6 cm NGM plates supplemented with carbenicillin and grown for 1–2 days at room temperature, and 80 μl 0.1M IPTG (Promega, Sunnyvale, CA) added to bacterial lawn to induce dsRNA expression. Eggs (whole-life RNAi) or adults (adult-only RNAi) were transferred to plates. RNAi of *bli-3*, *bli-4*, *elt-2*, *gfp*, *lin-26*, *pept-1*, *rab-7*, and *unc-112* was

performed from hatching, and *lgg-1, daf-16,* and *atg-18* RNAi used in lifespan analyses was performed from Day 1 of adulthood.

To confirm efficacy of *atg-18* RNAi in *daf-2* animals, transgenic WT or *daf-2(e1370)* animals expressing *atg-18p::atg-18::mCherry* from and extrachromosomal array were fed bacteria expressing *atg-18* dsRNA from hatching and imaged using a Leica fluorescence dissecting microscope at 23x magnification. Images were acquired using a Leica DFC310 FX camera with 4 second exposure. Fluorescence intensity in the anterior intestine was determined using Image J.

## Construction of transgenic strains

To construct the *lgg-1p::mcherry::gfp::lgg-1* vector, *mCherry* was amplified by PCR from a Clontech pmCherry expression vector and the PCR fragment was inserted into TOPO TA vector. *mCherry* was verified by sequencing and then removed using Kpn1 and inserted into an *lgg-1p::gfp:lgg-1* vector (*Meléndez et al., 2003*) upstream of *gfp*. The final product *lgg-1p::mcherry::gfp::lgg-1* (pMH878) was verified by sequencing.

An expression plasmid for neuronal-specific *rgef-1p::mCherry::gfp::lgg-1* was generated by Gateway cloning technology lambda (Thermo Fisher Scientific, Waltham, MA) (*Hartley et al., 2000*) using pDONR P4-P1R-*rgef-1p* (~1.6 kb, amplified from a vector provided by the Dillin lab), pDONR221-*mCherry::gfp::lgg-1* open-reading frame (amplified from pMH878), and pDONR-P2RP3-*unc-54 3'UTR* (amplified from pGH8 [*Frøkjaer-Jensen et al., 2008*]). The final product *rgef-1p::mcherry::gfp::lgg-1* (pMH1130) was verified by sequencing. An expression plasmid for *atg-18p::atg-18::mCherry* was similarly made using pDONR P4-P1R-*atg-18p* (540 bp, amplified from genomic DNA) and pDONR221-*atg-18* open-reading frame (amplified from genomic DNA). The final product *atg-18p::atg-18::mCherry* (pMH791) was verified by sequencing.

Plasmid DNA was prepared using a Mini or Midi Prep kit (Invitrogen, Waltham, MA or Qiagen, Hilden, Germany). Transgenic animals expressing an extrachromosomal array were created by gonadal microinjection of pMH878/*lgg-1p::mCherry::gfp::lgg-1,* pMH1130/*rgef-1p::mCherry::gfp::lgg-1,* or pMH791/*atg-18p::atg-18::mCherry* plus pRF4/*rol-6* co-injection marker into N2-Hansen animals. A list of strains made in this study is provided in *Supplementary file 2*. Integration of *lgg-1p::mCherry::gfp::lgg-1* was subsequently performed by γ-irradiation followed by outcrossing four times to N2-Hansen WT animals.

To show that *lgg-1p::mCherry::gfp::lgg-1* was functional and could rescue the *lgg-1* mutant embryonic lethality, MAH215/*lgg-1p::mCherry::gfp::lgg-1* was crossed to FX3489/*lgg-1(tm3489) II/+*, and the mCherry/GFP-positive F2 progeny were assayed for the homozygous *lgg-1* mutation by PCR. Mutated *lgg-1* is ~800 bp and WT *lgg-1* is ~1000 bp. Viable and fertile animals that were positive for mCherry/GFP were identified with mutant *lgg-1*. Primer information is available in *Supplementary file 3*.

## Quantification of autophagic vesicles

*C. elegans* were mounted live on a 2% agarose pad in M9 medium containing 0.1% NaN₃ and imaged using an LSM Zeiss 710 scanning confocal microscope, Z-stack images were acquired at 0.6 μm slice intervals at 63x. GFP excitation/emission was set to 493/517 nm to eliminate background autofluorescence. GFP::LGG-1 (GFP) or mCherry::GFP::LGG-1 (mCherry/GFP or mCherry only)-positive punctae were counted in the hypodermal seam cells, intestine, body-wall muscle, pharynx, and nerve-ring neurons in one 0.6 μm slice. The Z-position was selected where the nucleus could be clearly seen (for hypodermal seam cells and intestine), where striation could be seen (body-wall muscle), and where the lumen of the pharyngeal bulbs was in focus (for the pharynx and nerve-ring neurons). Punctae were quantified from images as follows: for body-wall muscle, punctae in one 1000 μm² area per 0.6 μm slice per animal, for the nerve-ring neurons total punctae between the pharyngeal bulbs in one 0.6 μm slice per animal, for the pharynx the number of punctae in the posterior pharyngeal bulb, and for the intestine and hypodermal seam cells, the number of punctae per cell per 0.6 μm slice. The number of mCherry-only punctae was calculated as (the total number of mCherry-positive punctae – the number of GFP-positive punctae). Statistical analysis of punctae was performed using Poisson regression calculated with the *R Core Team (2015)*. The intensity of red fluorescence compared to green is stronger and in order to see mCherry-positive punctae clearly, the gain of the red channel was purposely set lower. If the intensity of red fluorescence was

increased (which would overexpose the red punctae), cytoplasmic mCherry::GFP::LGG-1 appeared yellow (data not shown). For GFP::LGG-1(G116A) experiments, punctae in hypodermal seam cells were counted using a Zeiss Imager Z1, while at the microscope. The total number of punctae was counted in all visible hypodermal seam cells, whereas punctae quantification in the intestine, body-wall muscle, and pharynx was done as specified above. Imaging was performed using Zeiss Imager Z1 including apotome.2 at 100x magnification with a Hamamastsu orca flash 4LT camera and Zen 2.3 software. The average and SEM were calculated and data were analyzed using one-way analysis of variance (ANOVA) or two-way ANOVA as applicable (GraphPad Prism, La Jolla, CA).

## Bafilomycin A and LysoTracker treatment

Bafilomycin A (BafA; BioViotica, Dransfeld, Germany) was injected into the body cavity of *C. elegans* as previously described (*Wilkinson et al., 2015*). Briefly, 50 μM of BafA in 0.2% DMSO (25 mM stock in DMSO) or 0.2% DMSO (control) was co-injected with 2.5 ug/ml Texas Red Dextran or Cascade Blue Dextran, 3,000 Daltons (re-suspended in water at a stock concentration of 25 ug/ml; Thermo Fisher Scientific, Walltham, MA) into the anterior body cavity, intestine, or intestinal lumen, and the animals were allowed to recover for two hours before they were imaged by confocal microscopy. Injection of BafA could result in variability of LGG-1 punctae counts; however, injection of the 3,000 Dalton Texas Red Dextran alone (i.e. significantly larger than BafA) into ~100 worms showed highly reproducible Texas Red signal in the head region where LGG-1 punctae were quantified (data not shown), and the standard error of the mean was similar in DMSO- and in BafA-treated animals in all experiments performed (as well as in published work, see [*Wilkinson et al., 2015*]). Thus, notable variability in number of punctae is likely due to variability per animal rather than variable injection of BafA.

C. elegans were grown from egg to adulthood on *E. coli* OP50-seeded NGM plates containing 25 μM LysoTracker Deep Red (Thermo Fisher Scientific, Waltham, MA, mixed into plate media during prep) or an equivalent volume of DMSO (control) and subsequently imaged using confocal microscopy.

## Lifespan analysis

Lifespan was measured at 20°C as previously described (*Hansen et al., 2005*). Briefly, synchronized animals were transferred onto *E. coli* OP50-seeded plates and were raised at 20°C for WT/*daf*-2 lifespans while animals for WT/*glp-1* lifespans were left at 20°C for 24 hr then moved to 25°C for 48 hr. At Day 1 of adulthood all animals were moved to 20°C for the rest of their lives. The animals were then transferred to plates seeded with HT115 RNAi clones (adult-only RNAi treatment) or bacteria contain empty vector (controls) and scored every 1–3 days, as previously described (*Hansen et al., 2005*). Animals were scored as dead if they failed to respond to gentle prodding with a platinum wire pick. Censoring occurred if animals desiccated on the edge of the plate, escaped, ruptured, or suffered from internal hatching. Kaplan–Meier survival curves were constructed and statistical analysis was performed using STATA software (StataCorp, College Station, TX). p-values were calculated using the log-rank (Mantel–Cox) method.

## Analysis of the *sid-1* transgene

Intestine-specific *sid-1* transgene expression was assessed by PCR (see *Supplementary file 3* for primer list). All *daf-2* and *glp-1* strains carrying *sid-1* transgenes were tested and confirmed for RNAi efficiency using multiple tissue-specific RNAi clones for genes expressed in the intestine, body-wall muscle or hypodermis (*Figure 6—figure supplement 1A–B*, data not shown) as previously described for WT tissue-specific RNAi strains (*Figure 6—figure supplement 1*, and *Kumsta and Hansen, 2012*, and data not shown). Briefly, strains carrying *sid-1* transgenes were subjected to whole-life tissue-specific RNAi and imaged at Day 3 of adulthood. Worms were imaged in M9 medium containing 0.1% NaN$_3$ on a Leica fluorescence dissecting microscope at 8x magnification. Images were acquired using a Leica DFC310 FX camera with 2-second exposure.

## Immunoblotting

Total protein was extracted from 50 handpicked Day 1 adult animals grown on OP50 bacteria at 20°C. Animals were washed thoroughly in M9 buffer and centrifuged, and the pellets were lysed in 10 μl of

6x SDS sample buffer. The entire extract was separated by 4–20% SDS-PAGE (Thermo Fisher Scientific, Waltham, MA) and transferred to a PVDF membrane (Millipore, Hayward, CA). Immunoblotting was performed using primary anti-GFP (diluted 1:1000; Santa Cruz Biotechnology, Dallas, TX), anti-mCherry (diluted 1:500; Clontech, Mountain View, CA), and anti-LGG-1 (diluted 1:1000; sample kindly obtained from Abgent, San Diego, CA) antibodies and secondary horseradish peroxidase-conjugated goat anti-mouse (diluted 1:2000; Santa Cruz Biotechnology, Dallas, TX) or goat anti-rabbit (1:2000; Cell Signaling Technology, Danvers, MA) secondary antibodies. Immunoblots were developed using enhanced chemiluminescent reagent (Thermo Fisher Scientific, Waltham, MA).

### Immunofluorescence in *C. elegans* intestines

Day 1 adult transgenic animals expressing *mcherry::gfp::lgg-1* were picked into a drop of M9 on a glass slide and dissected by pulling the head and tail apart with forceps, resulting in the intestine popping out of the animal. Dissected intestines were transferred to a 1.5-ml tube and fixed in 4% paraformaldehyde in 0.1% PBS-Tween20 for one hour at room temperature, and rinsed at least three times in 0.1% PBS-Tween20. Intestines were blocked in 5% FBS +0.1% PBS-Tween20 overnight at 4°C. Primary antibodies were mouse anti-GFP (diluted 1:100; Santa Cruz Biotechnology, Dallas, TX), mouse anti-mCherry (diluted 1:50; Clontech, Mountain View, CA), and rabbit anti-LGG-1 (diluted 1:100; sample kindly providd by Abgent, San Diego, CA); they were diluted in block and incubated overnight at 4°C. Secondary antibodies used were goat anti-mouse AlexaFluor 546 (1:500 Life Technologies, Carlsbad, CA) and goat anti-rabbit AlexaFluor 488 (1:500 Life Technologies, Carlsbad, CA) and incubated in block overnight at 4°C. Dissected intestines were flat mounted in glycerol directly onto glass slides and imaged using an LSM Zeiss 710 scanning confocal microscope. Secondary antibody-only treatment did not have any specific antibody staining in the intestine (data not shown). Fixation quenched mCherry/GFP fluorescence in the intestine, but not in the pharynx, body-wall muscle or hypodermis, as these tissues were not exposed as much as the intestine to the fixing solution and sometimes appeared to be fluorescent (both green and red) following secondary antibody-only treatment. N2/WT animals treated with secondary antibodies only did not show fluorescence staining in the body-wall muscle, pharynx, or hypodermis demonstrating that fluorescence in transgenic animals is due to the reporter that was not quenched during fixation (data not shown).

## Acknowledgements

We thank members of the Hansen lab for insightful discussions, Dr. Alex Soukas for experimental advice and comments on the manuscript, Dr. Yongzhi Yang for strain construction (MAH679 and MAH680) and help characterizing these strains, Dr. Xingyu She for plasmid construction (pMH1130), Dr. Sara Gelino for strain construction (MAH113 and MAH134), and Chu-Chiao (Joyce) Chu for integrating and outcrossing MAH167. We also thank the Mitani lab for the FX3489 strain, Dr. Andrew Dillin for the AGD801 strain, Dr. Renaud Legouis for the RD202 strain, and Abgent for providing LGG-1 antibody sample. Statistical analysis of autophagosome and autolysosome quantification was performed by Dr. Xiayu Stacy Huang, Bioinformatics Core, SBP Institute. This work was funded by NIH/NIA grants AG038664 and AG039756 to MH, who is also supported by a Julie Martin Mid-Career Award in Aging Research from The Ellison Medical Foundation/AFAR.

## Additional information

### Funding

| Funder | Grant reference number | Author |
|---|---|---|
| National Institute on Aging | AG039756 | Malene Hansen |
| National Institute on Aging | AG038664 | Malene Hansen |
| American Federation for Aging Research | Mid-Career Award in Aging Research, M14329 | Malene Hansen |
| Ellison Medical Foundation | Mid-Career Award in Aging Research, M14329 | Malene Hansen |

The funders had no role in study design, data collection and interpretation, or the decision to submit the work for publication.

## Author contributions

JTC, Designed and performed all experiments, except the GFP::LGG-1(G116A) analysis, analyzed the data, created new strains, and wrote the manuscript; CK, Designed and performed the GFP::LGG-1(G116A) analysis, and analyzed the data; ABH, Made and injected the mCherry::GFP::LGG-1 construct; LMA, Performed a repeat of the lifespan assays involving knockdown of *daf-16*/Foxo, and analyzed the data; MH, Designed all experiments and wrote the manuscript

## Author ORCIDs

Malene Hansen, http://orcid.org/0000-0002-6828-8640

# Additional files

## Supplementary files

• Supplementary file 1. (A–B) Lifespan analysis of *daf-2* and *glp-1* animals subjected to whole-body or tissue-specific RNAi of autophagy genes during adulthood. Lifespan analysis of *glp-1(e2141)* and *glp-1(e2141); sid-1(qt9)* (A), and *daf-2(e1370)* and *daf-2(e1370); sid-1(qt9)* (B) animals carrying tissue-specific arrays to re-establish RNAi by expression of *sid-1* transgene from intestinal-specific (*vha-6*) or muscle-specific (*myo-3*) promoters. WT and *daf-2* animals were incubated at 20°C and fed from Day 1 of adulthood with bacteria containing empty vector (control) or expressing dsRNA encoding autophagy genes *atg-18/Wipi*, *lgg-1/Atg8* or *daf-16/Foxo*. *glp-1(e2141)* animals were raised at 25°C and moved to 20°C at Day 1 of adulthood for the rest of their lifespan. Individual experiments are numbered (Exp#). We note that *lgg-1* RNAi did not significantly shorten the lifespan of *glp-1(e2141); sid-1(qt9); vha-6p::sid-1* transgenic animals, but this treatment had relatively weak effects in *glp-1(e2141)* single mutants analyzed in parallel experiments. Moreover, the *glp-1; sid-1; vha-6p::sid-1* strain was, for unexplainable reasons, short-lived in experiment number 5. Whole-body, muscle-specific, and intestinal-specific inhibition of autophagy genes generally had no effect or slightly decreased the lifespan of wild-type N2 animals (JTC and MH, unpublished results). Data show the average lifespan (avg. LS), the number of events (# animals/Total; number of dead animals/total number of animals analyzed, the percentage lifespan extension (% change) of animals subjected to gene-specific vs control RNAi, and the p value for this comparison. p values were calculated using the Mantel–Cox log-rank test. *data shown in *Figure 6*.

• Supplementary file 2. *C. elegans* strains used in this study.

• Supplementary file 3. Primers used in this study.

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
