## [Decision Letter]

Thank you for submitting your article "Spatiotemporal regulation of autophagy during *C. elegans* aging" for consideration by *eLife*. Your article has been favorably evaluated by Sean Morrison (Senior Editor) and three reviewers, one of whom is a member of our Board of Reviewing Editors. The reviewers have opted to remain anonymous.

The reviewers have discussed the reviews with one another and the Reviewing Editor has drafted this decision to help you prepare a revised submission.

Summary:

In this manuscript, the authors investigated autophagic flux in multiple tissues during the aging process using the tandem-tagged Atg8/LGG-1 reporter. They found that autophagic activity declines with age in all major tissues. They also showed that long-lived *daf-2* and *glp-1* mutant animals exhibit unique age- and tissue-specific differences in autophagic activity. Compared with wild type animals, *daf-2* mutants display elevated autophagic activity, while *glp-1* mutants show reduced autophagic activity in the hypodermal seam cells. Using tissue-specific RNAi, the authors demonstrated that intestinal autophagy is essential for lifespan extension only in *glp-1* mutants. Overall, the work is interesting. The reviewers found that additional work is required to strengthen the manuscript to meet the standards for *eLife*.

Essential revisions:

1) This study relies heavily on the tandem nCherry-gfp-lgg-1 reporter to monitor autophagic flux. It has been shown that overexpressed Atg8/LC3 reporters not only form aggregates but can also label non-autophagic structures (such as the Golgi) in yeast and mammalian cells. Further validation of the tandem reporter using Atg gene mutant backgrounds would be necessary in the different tissues.

A) LGG-1/ATG8 must be lipidated in order for it to form punctate structures. The transgenic reporter could be crossed into various autophagy gene mutants that abolish LGG-1 lipidation, including *atg-4.1; atg-4.2, atg-16.1; atg-16.2, atg-3, atg7*, and then the formation of puncta can be examined. If mCherry-gfp-lgg-1 puncta are absent in those mutants, it indicates that the puncta represent autophagic structures. The expression of this tandem reporter could also be examined in mutants of the Atg1 kinase complex, in which LGG-1 displays a unique distribution pattern at embryonic stages.

B) Endogenous LGG-1, detected by anti-LGG-1, could be used to demonstrate the change in autophagy activity in aged animals.

2) The authors show that autophagy in hypodermal seam cells is induced in daf-2 mutants but blocked in germline-less *glp-1* mutants. However, the numbers of APs and ALs are pretty low in this tissue in wild type and mutant animals with or without Bafilomycin treatment. Other assays such as degradation of the well-characterized autophagy substrate SQST-1 in partial loss-of-function autophagy mutants need be performed to demonstrate the differential autophagy activity in *daf-2* and *glp-1* mutants.

3) Bafilomycin treatment was used to block lysosome degradation in this study. However, Bafilomycin was administered by injection, which may result in variation among animals. Genetic mutants with impaired lysosomal function, including *cup-5* and *laat-1*, could be used to follow the number of APs and ALs in aged animals or long-lived mutants.

[Editors' note: further revisions were requested prior to acceptance, as described below.]

Thank you for resubmitting your work entitled "Spatiotemporal regulation of autophagy during *C. elegans* aging" for further consideration at *eLife*. Your revised article has been favorably evaluated by Sean Morrison (Senior Editor) and three reviewers, one of whom is a member of our Board of Reviewing Editors.

The manuscript has been improved but there are some remaining issues that need to be addressed before acceptance, as outlined below:

Other assays are needed to demonstrate that mCherry::GFP::LGG-1 puncta are autophagic structures but not protein aggregates. Anti-LGG-1 antibody could be used to detect endogenous LGG-1 (not in animals carrying the transgene) pattern in larvae. SQST-1 could also be stained to see whether it is colocalized with mCherry::GFP::LGG-1 puncta (three color staining).

*Reviewer #1:*

The authors have addressed many of my concerns. But they need clarify/address the following remaining concerns:

1) *atg-3(bp412)* is not a null allele. Genetic nulls for two conjugation systems are lethal. Thus, alternative explanation for the formation of LGG-1 puncta in *atg-3(bp412)* larvae need be discussed.

2) The authors performed immunostaining against endogenous LGG-1 in animals carrying the mCherry::gfp::lgg-1 transgene. The antibody for sure will recognize MCHERRY::GFP::LGG-1 expressed from the transgene. Such experiments should be carried out in animals without carrying the transgene.

*Reviewer #2:*

In their revised manuscript, the authors test the punctate distribution of their new mCherry::GFP::LGG-1 reporter in various Atg loss-of-function backgrounds to validate its use for the analysis of autophagic activity. As expected, they show that the number of mCherry-positive, GFP-negative dots (autolysosomes) decreases in these situations. However, surprisingly, the number of structures positive for both mCherry and GFP increases upon loss of Atg genes such as Atg3 that is required for LGG-1 lipidation and membrane association. The authors consider these dots as autophagosomes, and argue that their data agree with the findings of a recent Science paper by the Mizushima lab. However, this is not true: the abnormal autophagosome-like structures forming in ATG conjugation gene knockout mammalian cells are negative for LC3 (the homolog of LGG-1), as LC3 does not associate with autophagic structures in the absence of its lipidation. See Figure 4H in Tsuboyama et al., 2016. It is firmly established that Atg8/LGG-1/LC3 lipidation is required for its membrane association in all tested models including yeast, worms and *Drosophila*, not only in mammals. I thus think that an unknown and variable proportion of the mCherry::GFP::LGG-1 dots observed in the various genetic backgrounds represent aggregates of the reporter protein (probably most of it is aggregates in Atg3 mutants), which undermines the validity of the whole study. This is especially true as the data are not verified with a robust independent test, such as endogenous LGG-1 puncta formation – it is insufficient to show that the overexpressed reporter is positive for anti-LGG-1! The main message of autophagy guidelines papers is to always rely on multiple independent tests when evaluating autophagy, so the revised manuscript still does not meet this fundamental requirement. The authors even cite Klionsky et al. 2016 Autophagy (in which the senior author is a co-author) representing a consensus recommendation of 2467 researchers in the field, and Dr. Hansen should also follow these guidelines. Taken together, the manuscript is solely based on a reporter that I do not think is reliable, so I am strongly against the publication of this preliminary work in any journal.

(Reviewer #2 corrected his original criticism after he read comments from the other reviewers. Reviewer #2 thought *atg-3(bp412)* is a genetic null, which led him to conclude that mCherry-GFP-LGG-1 puncta were protein aggregates. Actually, *atg-3(bp412)* is a hypomorphic allele. Genetic nulls for the two conjugation systems in *C. elegans* cause lethality and cannot be tested for this purpose.

This study contains several very interesting points, especially tissue-specific requirement for the longevity conferred by loss of function of *glp-1* and *daf-2*. So my (Reviewing Editor) suggestion is that the authors need test the validity of the reporter. If the authors have the antibodies, those experiments are easy to do.)

*Reviewer #3:*

The authors have made notable efforts to sufficiently address the issues raised by the reviewers. I have only one remaining issue.

Figure 7 is not necessary as the panels do not really help readers understand their conclusions. The panels are also misleading as they are not actual data but look oversimplified and biased versions of data. I suggest that the authors instead need to draw models that represent their conclusions of the manuscript.

---

## [Author Response]

*1) This study relies heavily on the tandem nCherry-gfp-lgg-1 reporter to monitor autophagic flux. It has been shown that overexpressed Atg8/LC3 reporters not only form aggregates but can also label non-autophagic structures (such as the Golgi) in yeast and mammalian cells. Further validation of the tandem reporter using Atg gene mutant backgrounds would be necessary in the different tissues.*

*A) LGG-1/ATG8 must be lipidated in order for it to form punctate structures. The transgenic reporter could be crossed into various autophagy gene mutants that abolish LGG-1 lipidation, including atg-4.1; atg-4.2, atg-16.1; atg-16.2, atg-3, atg7, and then the formation of puncta can be examined. If mCherry-gfp-lgg-1 puncta are absent in those mutants, it indicates that the puncta represent autophagic structures. The expression of this tandem reporter could also be examined in mutants of the Atg1 kinase complex, in which LGG-1 displays a unique distribution pattern at embryonic stages.*

To further validate our new mCherry::GFP::LGG-1 reporter, we have expressed and examined it in *atg-3(bp412)* and *atg-18(gk378)* mutants (we note that we also attempted to introduce the reporter in *atg-4.1(bp411)* and *unc-51(e369)* mutants, yet we were unable to optimize protocols and finalize these crosses in time for resubmission). In embryos, the *atg-3(bp412)* mutant has been reported to lack puncta as detected by using an antibody to label endogenous LGG-1 (Tian et al., 2010). Interestingly, we observed more autophagosomes (AP) and less autolysosomes (AL) in both *atg-3(bp412)* and in *atg-18(gk378)* mutants compared to wild-type (WT) transgenic animals (Figure 7). We also found that *atg-18(gk378)* mutants expressing GFP::LGG-1had increased number of GFP-positive punctae (i.e., APs; Figure 7
**–** unpublished results, C. Kumsta and M. Hansen). Similarly, we saw APs increase and ALs decrease in WT transgenic animals expressing mCherry::GFP::LGG-1 that were fed bacteria expressing dsRNA for two generations to inhibit *unc-51/atg-1* and *atg-13* (both part of the ATG1 kinase complex) in addition to *bec-1* and *atg-9* (Class II PI3K Complex – required for vesicle nucleation) and *atg-7* and *atg-18* (required for vesicle elongation to form a puncta) (Figure 8). Our data are consistent with recent work from Dr. Noburo Mizushima’s lab, which has examined Atg8 conjugation mutants affecting the closure (fission) step such as Atg5, Atg4b, and Atg3. Specifically, they observe that these mutants can form autophagosome-like structures that are labeled with the syntaxin protein Stx17 and can fuse with lysosomes but take ~10x longer to convert to an autolysosome compared to WT (Tsuboyama et al., Science, 2016). This may explain why we observed an increased numbers of APs and decreased ALs in *atg-3(bp412)* and *atg-18(gk378)* mutants expressing the mCherry::GFP::LGG-1 reporter. While we do not know the reason for the observed difference between embryonic and adult LGG-1 puncta numbers in *atg-3(bp412)* mutants, we speculate that it may relate to redundancy in adult functions of this gene.

Author response image 1.*atg-3* and *atg-18* mutants have more APs and less ALs compared to wild type.(**A**) Hypodermal seam cells in wild-type (WT), *atg-3(bp412)* and *atg-18(gk378)* transgenic animals expressing *mCherry::gfp::lgg-1,* imaged using confocal microscopy at Day 1 of adulthood. (B-C) Quantification of autophagosomes (AP) and autolysosomes (AL) in hypodermal seam cells (**B**) and intestine (**C**) of adult Day 1 WT, *atg-3(bp412)*, and *atg-18(gk378)* animals expressing *mCherry::gfp::lgg-1*. Data are the mean ± SEM of ≥29 animals combined from three experiments. ****p< 0.0001 by ANOVA. (**D**) Hypodermal seam cells in WT and *atg-18(gk378)* transgenic animals expressing *gfp::lgg-1* imaged using Zeiss apotome at Day 1 of adulthood. Data courtesy of Dr. Caroline Kumsta, Hansen lab.**DOI:**
http://dx.doi.org/10.7554/eLife.18459.021

Author response image 2.Quantification of autophagosomes and autolysosomes following inhibition of early autophagy genes using RNAi.(**A**, **B**) Quantification of autophagosomes (AP) and autolysosomes (AL) in hypodermal seam cells (**A**) and intestine (**B**) of adult Day 1 wild-type animals expressing *mCherry::gfp::lgg-1* fed for two generations with bacteria expressing control (empty vector) or *atg-1, atg-7, atg-9, atg-13, atg-18* or *bec-1* dsRNA. Data are the mean ± SEM of ≥28 animals combined from three experiments. ****p< 0.0001, **p<0.005, * p<0.05 to WT control by ANOVA.**DOI:**
http://dx.doi.org/10.7554/eLife.18459.022

*B) Endogenous LGG-1, detected by anti-LGG-1, could be used to demonstrate the change in autophagy activity in aged animals.*

We agree with the reviewer that detection of endogenous LGG-1 would be a useful way to further validate the new tandem reporter. We recently developed a new LGG-1 antibody that we have so far only used for Western blotting (Figure 1—figure supplement 1). We have now optimized the use of this antibody for immunofluorescence in dissected intestines of adult animals. Using our protocol/antibody, we have determined that GFP and mCherry both co-localize with anti-LGG-1, confirming that the tagged version is labeling LGG-1 positive puncta, at least in the intestine (new Figure 1—figure supplement 1). We have discussed these new data, where we say:

“Moreover, immunofluorescence analysis of the intestine of mCherry::GFP::LGG-1-expressing animals showed that both GFP- and mCherry-positive punctae co-localized with structures stained by an LGG-1 antibody (Figure 1—figure supplement 1).”

We also describe these data in the figure legend of Figure 1—figure supplement 1 where we say:

“(D-E) Immunofluorescence to detect endogenous LGG-1 (green) and GFP (D; red) or mCherry (E; red) in dissected intestines of wild-type (WT) animals. Data representative of ≥ 2 experiments (N ≥ 5 animals in each). Scale bars = 20 µM.”

Additionally, we describe our antibody immunofluorescence protocol in the Methods section, where we say:

“*Immunofluorescence of C. elegans intestines*

Day 1 adult transgenic animals expressing *mcherry::gfp::lgg-1* were picked into a drop of M9 on a glass slide and dissected by pulling the head and tail apart with forceps resulting in the intestine popping out of the animal. […] N2/WT animals treated with secondary antibodies only did not show fluorescence staining in the muscle, pharynx or hypodermis demonstrating that fluorescence in transgenic animals is due to the reporter that wasn’t quenched during fixation (data not shown).”

While we agree that this technique could be useful to demonstrate changes in autophagy in aged animals, the optimization of this new protocol/antibody proved challenging and time consuming and would have constituted a huge undertaking. Instead, we have stained Day 1 *daf-2* mutants, which appeared to display an increased number of puncta in their intestines compared to WT animals (Figure 9), consistent with our detailed quantifications of the fluorescent tandem LGG-1 reporter. Since we have yet to develop the protocol to reliably count punctae from imaged intestine, we prefer to not include these data in the revised manuscript.

Author response image 3.mCherry and GFP co-localize with endogenous LGG-1 in transgenic animals expressing mCherry::gfp::lgg-1.(**A-B**) Immunofluorescence to detect endogenous LGG-1 (green, rabbit anti-LGG-1 antibody made by Abgent) and GFP (A; red, mouse anti-GFP from Santa Cruz Biotechnology) or mCherry (B; red, mouse anti-mCherry from Clonetech) in dissected intestines of wild-type (WT, top, data also in Figure 1—figure supplement 1) or *daf-2(e1370)* animals (bottom). Data representative of ≥2 experiments (N≥5 animals in each). Scale bars = 20 µM.**DOI:**
http://dx.doi.org/10.7554/eLife.18459.023

*2) The authors show that autophagy in hypodermal seam cells is induced in daf-2 mutants but blocked in germline-less glp-1 mutants. However, the numbers of APs and ALs are pretty low in this tissue in wild type and mutant animals with or without Bafilomycin treatment. Other assays such as degradation of the well-characterized autophagy substrate SQST-1 in partial loss-of-function autophagy mutants need be performed to demonstrate the differential autophagy activity in daf-2 and glp-1 mutants.*

It is indeed interesting, but perhaps not surprising that hypodermal seam cells have “lower” number of APs and ALs in both WT and mutant animals with or without BafA treatment compared to other tissues since their cell size is much smaller than the regions quantified in other tissues. That said, we agree with the reviewer that examination of SQST-1 punctae in WT animals versus *daf-2* and *glp-1* mutants would be informative. To address this point, we used a novel *sqst-1p::sqst-1::gfp* reporter (the expression profile of these animals were equivalent to a previously published *sqst-1p::sqst-1::gfp* strain from Hong Zhang’s lab, data not shown), which we crossed to *glp-1(e2141)* and *daf-2(e1370)* mutants. When we analyzed Day 1 animals in their head region where the expression of this reporter is most prominent in adult animals (Figure 10), we found that *glp-1* mutants had similar numbers of SQST-1 punctae to WT animals, whereas *daf-2* animals had fewer punctae compared to WT animals (Figure 10). BafA treatments potently increased punctae in *daf-2* animals, consistent with enhanced autophagic activity between the two pharyngeal bulbs in *daf-2* mutants (Figure 10). Similarly, BafA treatment also increased punctae in *glp-1* mutants compared to WT. However, this region likely consists mostly of neurons, and any hypodermal seam cells are difficult to see and distinguish from the ventral/dorsal nerve cord in this strain (Figure 10). Thus, we would like to refrain from making any conclusions in regards to the hypodermal seam cells with this novel SQST-1 reporter, and although we tried, we have not been successful in making a useful reporter expressing SQST-1 from hypodermal promoters for this resubmission.

Instead, we have expanded our analysis of seam cells by analyzing *daf-2* and *glp-1* mutants in which we inhibited autophagy by *rab-7* or control (empty vector) RNAi from hatching. Consistent with active autophagy in the hypodermal seam cells of *daf-2* mutants, inhibition of *rab-7* resulted in an further increase in APs and ALs, while *glp-1* mutants showed no change in APs and ALs following *rab-7* inhibition (included as new Figure 2—figure supplement 3), suggesting a block of autophagy as our original BafA data indicated. We have inserted a comment about this additional experiment, where we say:

“We observed similar results when we blocked autophagy by *rab-7* RNAi (Figure 2—figure supplement 3).”

Author response image 4.The number of SQST-1/p62 puncta is decreased in *daf-2* animals compared to wild type.(**A**, **B**) Whole-body expression of new *sqst-1p::sqst-1::gfp* reporter in a wild-type (WT) animal at Day 1 of adulthood. Scale bar = 200 µm. Overlay of GFP and bright-field images (**A**) and GFP only (**B**). (**C**, **D**) Images of the head region quantified (between pharyngeal bulbs indicated by dotted lines) in WT animals treated with DMSO (**C**) or BafA (**D**). AB = anterior bulb; TB = terminal bulb. Scale bar = 20 µM. (**E**, **F**) Quantification of the number of SQST-1::GFP puncta in *daf-2(e1470)* and WT animals raised at 20^o^C (**E**) or *glp-1(e2141)* and WT animals raised at 25^o^C (**F**) and injected with DMSO or BafA at Day 1 of adulthood. Data are the mean ± SEM of ≥30 animals combined from three experiments. ****p< 0.0001, **p<0.005, * p<0.05 by ANOVA.**DOI:**
http://dx.doi.org/10.7554/eLife.18459.024

*3) Bafilomycin treatment was used to block lysosome degradation in this study. However, Bafilomycin was administered by injection, which may result in variation among animals.*

We are aware that such variability could exist in the Bafilomycin A (BafA) injection assay (which we developed to minimize the time to reach steady-state conditions (2 hrs)). However, we argue that such variability does not significantly affect our interpretations for multiple reasons. First, we injected Texas Red Dextran (5000 Daltons, i.e., significantly larger than BafA) into a large number of worms (~100) to determine reproducibility of injection. In the head region, where we quantified punctae in all assays, we saw a highly reproducible Texas Red Dextran signal (Figure 11). Second, we note that the standard error of the mean is similar in DMSO and in BafA-treated animals in each of our many performed experiments (as well as in published work, see Wilkinson et al., Molecular Cell, 2015), suggesting that any variability in the number of punctae is likely due to variability per animal rather than variable injection of BafA. We have made a comment to this experimental point in the Methods, where we say:

“Injection of BafA could result in variability of LGG-1 punctae counts, however, injection of a Texas Red Dextran (5,000 Daltons, i.e., significantly larger than BafA) into ~100 worms showed highly reproducible Texas Red signal in the head region where LGG-1 punctae were quantified (data not shown), and the standard error of the mean is similar in DMSO- and in BafA-treated animals in all experiments performed (as well as in published work see (Wilkinson et al., 2015)). Thus, notable variability in number of punctae is likely due to variability per animal rather than variable injection of BafA.”

Author response image 5.Texas Red Dextran injections are very reproducible.(**A**, **B**) Injection of Texas Red Dextran (5000 Daltons) into wild-type (WT) animals. Images (8x magnification) from 2 experiments (A and B) with 50 injected animals analyzed 10 at a time (i.e., 5 panels of 10 animals) per experiment. Top row = Brightfield, middle row = 0.25 second exposure, bottom row = 1 second exposure. Scale bars = 200 µM.**DOI:**
http://dx.doi.org/10.7554/eLife.18459.025

*Genetic mutants with impaired lysosomal function, including cup-5 and laat-1, could be used to follow the number of APs and ALs in aged animals or long-lived mutants.*

Throughout the study, we have blocked autophagy by using adult-only autophagy gene RNAi and short-term (2 hour) BafA treatments to avoid contributions of developmental autophagy. Therefore using genetic mutants to assess lysosomal function in aged animals is not an ideal experiment (plus such mutants were not currently available through the *C. elegans* stock center). The best experiment would be to use adult-only RNAi, but it is difficult to induce changes in fluorescent LGG-1 reporters with adult-only RNAi treatments (Hansen et al., PLOS Genetics, 2008). That said, we agree with the reviewer that it would be interesting to further investigate the effects of blocking lysosomal function (noting that BafA was used to block lysosomal acidification). To this end, we fed bacteria expressing dsRNA for the two lysosomal genes *laat-1* and *cup-5* to animals expressing mCherry::GFP::LGG-1 for two generations, and analyzed these animals on Day 1 of adulthood. This analysis showed increases in APs (hypodermal seam cells and intestine) and decreases in ALs in the intestine and pharynx, but not in hypodermal seam cells and in muscle (Figure 12). While the differences between tissues at this point remain unclear, these data are overall consistent with a decrease in autophagic activity at Day 1 of adulthood following inhibition of *laat-1* and *cup-5.*

Author response image 6.Quantification of autophagosomes and autolysosomes following Inhibition of lysosomal genes *cup-5* and *laat-1*.(**A-D**) Quantification of autophagosomes (AP) and autolysosomes (AL) in hypodermal seam cells (**A**), intestine (**B**), pharynx (**C**), and muscle (**D**) of adult Day 1 wild-type (WT) animals expressing *mCherry::gfp::lgg-1* fed for two generations with bacteria expressing empty vector (control) or *cup-5* or *laat-1* dsRNA. Data are the mean ± SEM of ≥45 animals combined from three experiments. ****p< 0.0001, **p<0.005, *p<0.05 by ANOVA.**DOI:**
http://dx.doi.org/10.7554/eLife.18459.026

[Editors' note: further revisions were requested prior to acceptance, as described below.]

*The manuscript has been improved but there are some remaining issues that need to be addressed before acceptance, as outlined below:*

*Other assays are needed to demonstrate that mCherry::GFP::LGG-1 puncta are autophagic structures but not protein aggregates.*

We have made a number of experimental and textual changes to address the reviewer’s remaining concerns. While further validation of the tandem reporter is possible, yet technically challenging (see below), we have instead reorganized our revised manuscript to better emphasize data with the well-established GFP::LGG-1 reporter (new Figure 1). To further increase confidence in this reporter, we have analyzed a lipidation-deficient GFP::LGG-1(G116A) reporter (Manil-Segalen et al., Developmental Cell, 2014). Specifically, we show that no or few punctae are formed over time in the intestine, body-wall muscle, and pharynx of WT, *daf-2,* or *glp-1* animals expressing GFP::LGG-1(G116A) (new Figure 1—figure supplement 1), arguing that GFP::LGG-1 punctae represent autophagosomes (APs) and do not seem to form aggregates with age in these tissues.

Unexpectedly, we observed GFP::LGG-1(G116A) punctae in hypodermal seam cells of *glp-1* mutants (former Figure 4, now Figure 4—figure supplement 9E), indicating that GFP::LGG-1 may aggregate in this cell type. Likewise, we observed GFP::LGG-1(G116A) punctae in *atg-3(bp412)* and *atg-18(gk378)* autophagy mutants (newFigure 4—figure supplement 9H-I, and data not shown, see also response to reviewer #1 below), similar to observations in mammalian Atg5 knockout cells overexpressing GFP::Atg8/LC3 (Kuma *et al.*, Autophagy, 2007). Collectively, these findings emphasize the importance of using a GFP::LGG-1(G116A) reporter to help assess whether GFP::LGG-1 punctae are likely to represent APs in any given context. We have therefore excluded the *daf-16* data set presented in the Discussion of the previous manuscript (revision 1), while analysis of *daf-16;* GFP::LGG-1(G116A) mutants are underway. Overall, we appreciate the opportunity to carry out these important control experiments, as they constitute a significant advance in the field.

While we recognize the usefulness of a similar mCherry::GFP::LGG-1 lipidation mutant, we were unsuccessful in creating such a new reporter in the resubmission period. However, we remain confident that the new tandem-tagged reporter reliably monitors APs and autolysosomes (AL) in *C. elegans* for multiple reasons. Specifically, we show (1) mCherry::GFP::LGG-1 is full length (Figure 2—figure supplement 2), and (2) is able to rescue a lethal *lgg-1* null mutant (data not shown), that (3) AP numbers with and without BafA treatment are similar in essentially every tissue and stage of adulthood between the well-established GFP::LGG-1 reporter and the new mCherry::GFP::LGG-1 reporter (Figure 1, Figure 3–Figure 5 and Figure 3—figure supplement 8, Figure 4—figure supplement 94*daf-2,* and *glp-1* animals (Figure 2), that (5) inhibition of AP conversion by *rab-7* RNAi (Figure 2) decreases ALs in the intestine of Day 1 WT animals, that (6) a *cst-1/Stk4* hippo kinase mutant with blocked autophagy does not respond to BafA (Figure 2—figure supplement 5C), and finally, as shown previously (Figure 10, see previous response to reviewers), (7) we observed that Day 1 *daf-2* mutants had fewer SQST-1::GFP punctae compared to WT, consistent with increased autophagy and conclusions made using the new tandem reporter. Irrespective of these lines of evidence, we acknowledge that we cannot fully account for 15% red-only punctae (Figure 2). These punctae could represent aggregates, or, alternatively, inefficient Lysotracker staining (as noted in the original and current manuscript). While we at present cannot distinguish between these possibilities, this estimate does not significantly offset any of our conclusions, and we observe similar% red-only punctae between genotypes (new Figure 2—figure supplement 5D).

To fully recognize this limitation and that our validation of the tandem-reporter was primarily done in the intestine of young WT animals, we have revisited our conclusions so they are based on consistent results between the mCherry::GFP::LGG-1 and GFP::LGG-1 reporters, or with clear note of limitations if a possible conclusion is suggested from AL counts only. Lastly, we have expanded the Discussion to carefully discuss the limitations of the reagents used in our study, including the need to fully evaluate neurons with an LGG-1(G116A) reporter (as the existing reporter did not allow accurate identification of neurons, see figure legend to new Figure 1—figure supplement 1).

Our conclusions remain unchanged overall, except for *glp-1* hypodermal seam cells (as noted above), and the muscle analysis of young *glp-1* animals, in which the flux analysis of the GFP::LGG-1 and mCherry::GFP::LGG-1 reporters, for unknown reasons, gave variable results in this one case (Figure 5 and Figure 3—figure supplement 8F’). While these two cases represented novel examples of a potential autophagy block in a long-lived animal, we note that our data set still reports on an example of a block of autophagy in a long-lived mutant, namely in the pharynx of young *daf-2* mutants. To reflect this change, we no longer explicitly highlight potential autophagy blocks in the abstract, but discuss appropriately in the main text.

Overall, we conclude that our revised manuscript, despite its noted limitations, represents a significant advance of our ability to measure autophagy in a multi-cellular organism, as well as our understanding of the role of autophagy during aging, and it provides an important new tool for the *C. elegans* field to be further validated in the future.

*Anti-LGG-1 antibody could be used to detect endogenous LGG-1 (not in animals carrying the transgene) pattern in larvae.*

We would like to clarify that we carried out LGG-1 antibody staining in transgenic animals since it was important to verify that mCherry::GFP::LGG-1 positive punctae were positive for both fluorophores as well as LGG-1, i.e., to argue that full-length protein was present in these punctae. We have made this point clearer in the resubmitted manuscript by moving the comment to page 8 as an introductory comment. That said, we recognize that staining for endogenous LGG-1 in the intestine (the only tissue we are capable of performing immunohistochemistry in adult animals at this point) would be useful to confirm differences observed with the fluorescent LGG-1 reporters of WT and long-lived mutants (while noting that this approach cannot differentiate the AP and AL compartments, as a mCherry::GFP::LGG-1 reporter is expected to do, nor would it allow to conclude if a positive structure is an aggregate or a *bona fide* autophagic vesicle). During the resubmission period, we therefore set out to antibody stain non-transgenic N2 WT and long-lived mutants. For these efforts, we had to use a new batch of the LGG-1 antibody that we previously obtained from Abgent, but this reagent unexpectedly resulted in Western blots of much reduced quality (data not shown), and we have been unable to carry out additional immunohistochemistry experiments.

That said, we note that we indeed stained N2 WT animals in our LGG-1 immunostainings for the original submission, and observed LGG-1-positive punctae in these animals, as expected (Figure 13).

Author response image 7.Antibody staining in N2 (Wild Type) animals.(**A-B**) Day 1 WT animals stained with rabbit anti-LGG-1 (green middle) and either mouse anti-mCherry (**A**), or mouse anti-GFP (**B**) (which here functioned as negative controls). Secondary antibody control had no fluorescence in either green or red channels (data not shown). Left column is merged image DIC, anti-LGG-1 and anti-mCherry or anti-GFP, middle is anti-LGG-1 only (green), and right is either anti-mCherry (**A**) or anti-GFP (**B**). Data representative of one experiment (N≥5 animals in each condition).**DOI:**
http://dx.doi.org/10.7554/eLife.18459.027

*SQST-1 could also be stained to see whether it is colocalized with mCherry::GFP::LGG-1 puncta (three color staining).*

We thank the editor for this suggestion, but we do not have access to an SQST-1 antibody. While an interesting experiment, it may, however, not be fully conclusive for the following reason. We have done co-localization assays of an unpublished TOMATO::LGG-1 strain from our lab and SQST-1::GFP (HZ589). While we observed frequent co-localization, we detected multiple examples of punctae positive for SQST-1 but not LGG-1, and vice versa (CK and MH, unpublished results). This is to be expected, since LGG-1 could be recruiting different adaptors, and SQST-1 may not always be targeted for autophagic degradation. Moreover, a co-localization event would not conclusively argue for autophagy as aggregated LGG-1 could potentially bind to SQST-1 even though it may not yet have been destined for autophagic degradation.

In conclusion, believe that the best experiment to do would be to co-stain with another membrane marker (i.e., Atg5 before closure, or Syntaxin-17 after closure of the autophagic vesicle), yet such reagents are, to our knowledge, unfortunately not currently available in the *C. elegans* field.

Reviewer #1:

*The authors have addressed many of my concerns. But they need clarify/address the following remaining concerns:*

*1) atg-3(bp412) is not a null allele. Genetic nulls for two conjugation systems are lethal. Thus, alternative explanation for the formation of LGG-1 puncta in atg-3(bp412) larvae need be discussed.*

We thank the reviewer for having us revisit the *atg-3(bp412)* mutant, which we showed in the previous response to the reviewers displayed increased AP and decreased AL numbers in hypodermal seam cells and in the intestine of young adults expressing the tandem reporter (Figure 7; similar observations were made with *atg-18(gk378)* and multiple autophagy gene RNAi clones in Figure 8, see previous response to reviewers). Since *atg-3(bp412)* (and the other conditions we tested) are hypomorphic, we reasoned that AP-like structures could still be formed but were delayed, similar to recent work from Dr. Mizushima’s lab (Tsuboyama et al., Science, 2016). However, as noted above, we have now created and analyzed *atg-3(bp412)* mutants expressing the GFP::LGG-1(G116A) lipidation mutant and find many punctate structures in these animals (new Figure 4—figure supplement 9H, and data not shown). We observed similar results when analyzing the *atg-18(gk368)* mutants (new Figure 4—figure supplement 9I, and data not shown). These observations instead argue for possible aggregation events in these mutants, similar to mammalian Atg5 knockout cells which accumulate GFP::Atg8/LC3 in punctate structures shown to be inclusion bodies (Kuma et al., Autophagy, 2007). As noted above, we have now included the hypodermal seam cell counts of these animals as new data in the manuscript,

*2) The authors performed immunostaining against endogenous LGG-1 in animals carrying the mCherry::gfp::lgg-1 transgene. The antibody for sure will recognize MCHERRY::GFP::LGG-1 expressed from the transgene. Such experiments should be carried out in animals without carrying the transgene.*

Please see response to editor above.

Reviewer #2:

*In their revised manuscript, the authors test the punctate distribution of their new mCherry::GFP::LGG-1 reporter in various Atg loss-of-function backgrounds to validate its use for the analysis of autophagic activity. As expected, they show that the number of mCherry-positive, GFP-negative dots (autolysosomes) decreases in these situations. However, surprisingly, the number of structures positive for both mCherry and GFP increases upon loss of Atg genes such as Atg3 that is required for LGG-1 lipidation and membrane association. The authors consider these dots as autophagosomes, and argue that their data agree with the findings of a recent Science paper by the Mizushima lab. However, this is not true: the abnormal autophagosome-like structures forming in ATG conjugation gene knockout mammalian cells are negative for LC3 (the homolog of LGG-1), as LC3 does not associate with autophagic structures in the absence of its lipidation. See Figure 4 in Tsuboyama et al., 2016. It is firmly established that Atg8/LGG-1/LC3 lipidation is required for its membrane association in all tested models including yeast, worms and Drosophila, not only in mammals. I thus think that an unknown and variable proportion of the mCherry::GFP::LGG-1 dots observed in the various genetic backgrounds represent aggregates of the reporter protein (probably most of it is aggregates in Atg3 mutants), which undermines the validity of the whole study.*

As stated in the response to editor above, we recognize the limitations of the new tandem-LGG-1 reporter, and have reorganized the manuscript to better emphasize the GFP::LGG-1 data set. Moreover, we have clearly discussed the need for further investigation of the mCherry::GFP::LGG-1 reporter, including validation of this reporter at older age, as well as analyses of G116A mutant versions expressed from endogenous and neuronal promoters (see Discussion of revised manuscript).

As also stated above, we show that no or few GFP punctae are formed over time in the major tissues of WT, *daf-2,* or *glp-1* animals expressing GFP:LGG-1(G116A) (new Figure 1—figure supplement 1), arguing that GFP::LGG-1 punctae represent autophagosomes (APs) and not aggregates. Unexpectedly, we observed GFP::LGG-1(G116A) punctae in hypodermal seam cells of *glp-1* mutants (former Figure 3, now Figure 4—figure supplement 9), indicating that GFP::LGG-1 may aggregate in this cell type. Likewise, we observed GFP::LGG-1(G116A) punctae in *atg-3(bp412)* and *atg-18(gk378)* autophagy mutants (new Figure 4—figure supplement 9, and data not shown), similar to observations in mammalian Atg5 knockout cells overexpressing GFP::Atg8/LC3 (Kuma et al., Autophagy, 2007). Collectively, these findings further validate the GFP::LGG-1 reporter and increase confidence in the GFP::LGG-1 data set, but also emphasize the use of a GFP::LGG-1(G116A) reporter to help assess whether GFP::LGG-1 punctae could represent APs in any given context.

While a similar mCherry::GFP::LGG-1 lipidation mutant will be important to create and analyze going forward, we note that our data set is overall very consistent between the GFP::LGG-1 reporter and the AP counts with the new mCherry::GFP::LGG-1 reporter. Regarding the latter, we agree that additional analysis of the red compartment is necessary, in particular since we cannot fully account for 15% red-only punctae (Figure 2), as also stated above. To this end, we have now also estimated this percentage in *daf-2* and *glp-1* animals (new Figure 2—figure supplement 5D), which show similar or smaller% red-only punctae in the intestine of Day 1 adults increasing confidence in our comparisons of the red channel/AL compartment between genotypes.

*This is especially true as the data are not verified with a robust independent test, such as endogenous LGG-1 puncta formation – it is insufficient to show that the overexpressed reporter is positive for anti-LGG-1!*

As noted in the response to the editor above, we initially carried out LGG-1 antibody staining in transgenic animals since it was important to verify that mCherry::GFP::LGG-1 positive punctae were positive for both mCherry, GFP and LGG-1, i.e., to argue that these punctae contained full-length protein.

*The main message of autophagy guidelines papers is to always rely on multiple independent tests when evaluating autophagy, so the revised manuscript still does not meet this fundamental requirement. The authors even cite Klionsky et al. 2016 Autophagy (in which the senior author is a co-author) representing a consensus recommendation of 2467 researchers in the field, and Dr. Hansen should also follow these guidelines. Taken together, the manuscript is solely based on a reporter that I do not think is reliable, so I am strongly against the publication of this preliminary work in any journal.*

We agree with the reviewer about the importance of using multiple approaches where at all possible to support claims regarding autophagy. For this study, we developed a new tandem-tagged mCherry::GFP:LGG-1 reporter, similar to Atg8 reagents widely used in mammalian cells, that, to our knowledge, have not previously been validated to the extent that we have validated it here in any system (see response to editor above). For this resubmission, we have reorganized our revised manuscript to better emphasize data obtained with the GFP::LGG-1 reporter (new Figure 1) and we further validated our GFP::LGG-1 data set by analyzing a GFP::LGG-1(G116A) lipidation mutant in all relevant settings (new Figure 1—figure supplement 1, Figure 4—figure supplement 9D-E, H-I, and data not shown), thus substantially increasing confidence in the GFP::LGG-1 data set. Thus, we have extensively validated the main reagents we used to carry out a comprehensive and unparalleled spatiotemporal analysis of autophagy in a multi-cellularorganism. As clearly noted in the manuscript, we agree that these observations should be confirmed by additional approaches, including by biochemical methodologies (which we are currently developing, but which represent long-term efforts). Indeed, such confirmatory efforts would require extensive assay development and optimization since no alternative assays are currently available to make comprehensive tissue- or whole organism autophagic activity assessments in *C. elegans* (besides the reporters recently developed in Dana Miller’s lab (Chapin et al., Aging, 2015); however, these, too, would require further validation, see Discussion for limitations of this reporter).

Still, we attempted to develop an additional assay in the resubmission period to further support our claims about age-related changes in autophagy in *C. elegans*. Specifically, we tried to implement a ‘free GFP cleavage’ assay using BafA soaking in adult animals followed by Western blotting, similar to previous protocols in mammalian cells (Klionsky et al., Autophagy, 2016). However, our results were unfortunately quite variable between repeats (Y. Yang and MH, unpublished results) and the assay needs further optimization, including finding the minimal saturating BafA concentrations for WT and long-lived mutants (Ni et al., Autophagy, 2016).

Overall, we conclude that we have done everything reasonably possible to validate the reagents we used in this study, and we would at this point prefer to share them and our comprehensive autophagy analysis during *C. elegans* aging with the research community, while clearly listing the limitations of our study.

*(Reviewer #2 corrected his original criticism after he read comments from the other reviewers. Reviewer #2 thought atg-3(bp412) is a genetic null, which led him to conclude that mCherry-GFP-LGG-1 puncta were protein aggregates. Actually, atg-3(bp412) is a hypomorphic allele. Genetic nulls for the two conjugation systems in C. elegans cause lethality and cannot be tested for this purpose.*

*This study contains several very interesting points, especially tissue-specific requirement for the longevity conferred by loss of function of glp-1 and daf-2. So my (Reviewing Editor) suggestion is that the authors need test the validity of the reporter. If the authors have the antibodies, those experiments are easy to do.)*

Please see response to editor above.

Reviewer #3:

*The authors have made notable efforts to sufficiently address the issues raised by the reviewers. I have only one remaining issue.*

*Figure 7 is not necessary as the panels do not really help readers understand their conclusions. The panels are also misleading as they are not actual data but look oversimplified and biased versions of data. I suggest that the authors instead need to draw models that represent their conclusions of the manuscript.*

We agree with the reviewer and have removed the model figure from the manuscript.